# Gut bacteria-derived sphingolipids alter innate immune responses to oral cholera vaccine antigens

Denise Chac [1], Frederick J. Heller[2], Hasan Al Banna [3,4], M. Hasanul Kaisar [3], Susan M. Markiewicz[1], Emily L. Pruitt [5], Fahima Chowdhury[3], Taufiqur R. Bhuiyan[3], Afroza Akter[3], Ashraful I. Khan[3], Mia G. Dumayas[1], Amelia Rice[1], Polash Chandra Karmakar[3], Pinki Dash [3], Regina C. LaRocque[6,7], Edward T. Ryan[6,7,8], Libin Xu [5], Samuel S. Minot [9], Jason B. Harris[10,11], Firdausi Qadri [3] & Ana A. Weil [1,12] ✉

The degree of protection conferred after receiving an oral cholera vaccine (OCV) varies based on age, prior exposure to *Vibrio cholerae*, and unknown factors. Recent evidence suggests that the microbiota may mediate some of the unexplained differences in oral vaccine responses. Here, we use metagenomic sequencing of the fecal microbiota at the time of vaccination and relate microbial features to immune responses after OCV using a reference-independent gene-level method. We find that the presence of sphingolipid-producing bacteria is associated with the development of protective immune responses after OCV. We test these associations by stimulating human macrophages with *Bacteroides xylanisolvens* metabolites and find that sphingolipid-containing extracts increase innate immune responses to OCV antigens. Our findings demonstrate a new analytic method for translating metagenomic sequencing data into strain-specific results associated with a biological outcome, and in validating this tool, we identify that microbe-derived sphingolipids impact immune responses to OCV antigens.

Cholera is a severe diarrheal disease caused by the bacterium *Vibrio cholerae*, and millions of cases occur annually[1]. Despite increased surveillance and ongoing efforts to provide safe water to vulnerable populations, there has been a recent re-emergence or new outbreaks of cholera in Haiti, Lebanon, and Syria, in addition to ongoing endemic spread in over 50 countries in Asia and Africa[2–4]. Oral cholera vaccines (OCV) are important tools in cholera control, but provide a short duration of protection and are less effective in children under five years of age[5].

Long-term immunologic protection from cholera is incompletely understood. The vibriocidal titer is a commonly used measure of the humoral response to *V. cholerae* infection, and although higher titers are associated with protection from disease, titers wane prior to loss of protective immunity[6,7]. Antibody responses to natural infection are

[1]Department of Medicine, University of Washington, Seattle, WA, USA. [2]Duke University School of Medicine, Durham, NC, USA. [3]Infectious Diseases Division, International Centre for Diarrheal Disease Research, Dhaka, Bangladesh. [4]Department of Chemistry and Biochemistry, University of Maryland, Baltimore, MD, USA. [5]Department of Medicinal Chemistry, University of Washington, Seattle, WA, USA. [6]Department of Medicine, Harvard Medical School, Boston, MA, USA. [7]Division of Infectious Diseases, Massachusetts General Hospital, Boston, MA, USA. [8]Department of Immunology and Infectious Diseases, Harvard School of Public Health, Boston, MA, USA. [9]Bioinformatics Core, Fred Hutchinson Cancer Research Center, Seattle, WA, USA. [10]Department of Pediatrics, Harvard Medical School, Boston, MA, USA. [11]Division of Global Health, MassGeneral Hospital for Children, Boston, MA, USA. [12]Department of Global Health, University of Washington, Seattle, WA, USA. ✉e-mail: anaweil@uw.edu

largely directed toward cholera toxin (CT) and the *V. cholerae* lipo-polysaccharide (LPS)[8]. In contrast with other toxin-mediated infections, CT-specific antibody responses are associated with only short-term protection from cholera[5,9,10]. The *V. cholerae* O1 serogroup is responsible for the majority of epidemic cholera, and serogroup specificity is defined by the composition of the O-specific polysaccharide (OSP) of the LPS. The O1 serogroup is subdivided into serotypes, and Inaba and Ogawa serotypes are the most dominant, and differ only by a single methyl group of the terminal sugar on the O-antigen[11]. After infection, OSP-specific memory B cells (MBCs) are detectable for up to one year[12], and upon exposure to *V. cholerae*, these cells are presumed to differentiate rapidly into antibody-secreting plasma cells to produce antibodies that act at the mucosal surface to protect from infection[9,13–16]. This is supported by longitudinal studies of household contacts of cholera patients, in which persons with elevated OSP-specific MBCs at the time of exposure to *V. cholerae* were protected from infection[9,17]. Recent in vivo studies established that OSP-specific antibodies reduce *V. cholerae* colonization in a motility-dependent manner, and thus, are thought to be a mediator of protection in human infection[18]. A collection of genetic, genomic, and laboratory-based studies has established that innate immune responses occur after acute infection and have an important role in shaping long-term memory B cell responses[13,16,19–22]. Innate immune responses contribute to the development and maintenance of OSP-specific MBCs that later reactivate by promoting upregulation of effector proteins and stimulating follicular helper T cell development for the provision of help to support B cell development, proliferation, and differentiation in germinal centers[22–24].

On a population level, vaccination with a whole-cell killed OCV stimulates an OSP-specific MBC response in half of OCV recipients[12,25–27]. While demographic factors such as age correlate with the likelihood of developing a protective vaccine response, biological mechanisms of these response differences have not been identified[26,28,29]. The concept that gut microbiota differences may mediate differences in host response to OCV is supported by associations between the microbiota and responses to other oral vaccines[30–33], the observation that immune responses to OCV were reduced in recipients with bacterial intestinal overgrowth[34], and our prior study that detected associations between the gut microbiota and OSP-specific MBC responses to OCV in a separate cohort[25]. To understand how gut microbes present at the time of vaccination could impact the host response to OCV, we applied a novel gene-level association method to metagenomic sequencing data from the microbiota at the time of OCV receipt. This analytic technique was selected and adapted to our analysis to identify specific microbial isolates associated with our outcome of interest, and this enabled us to experimentally test our sequencing study findings. We found that the presence of *Bacteroides* species capable of producing sphingolipids was correlated with the development of protective immune responses after OCV. To validate these findings, we exposed human macrophages to *Bacteroides*-derived sphingolipids in vitro and found a greater innate immune response to *V. cholerae* antigens, which suggests that gut microbial metabolites may impact immune responses to OCV.

## Results

### Immune responses to OCV are not correlated with microbiome composition

Eighty-nine participants received a whole-cell killed *V. cholerae* OCV with recombinant cholera toxin B and underwent measurements of post-vaccine immune response in Dhaka, Bangladesh (Fig. 1a)[27]. Stool collected on the day of OCV administration underwent DNA extraction and metagenomic sequencing. Twenty-two participants were aged 2-5

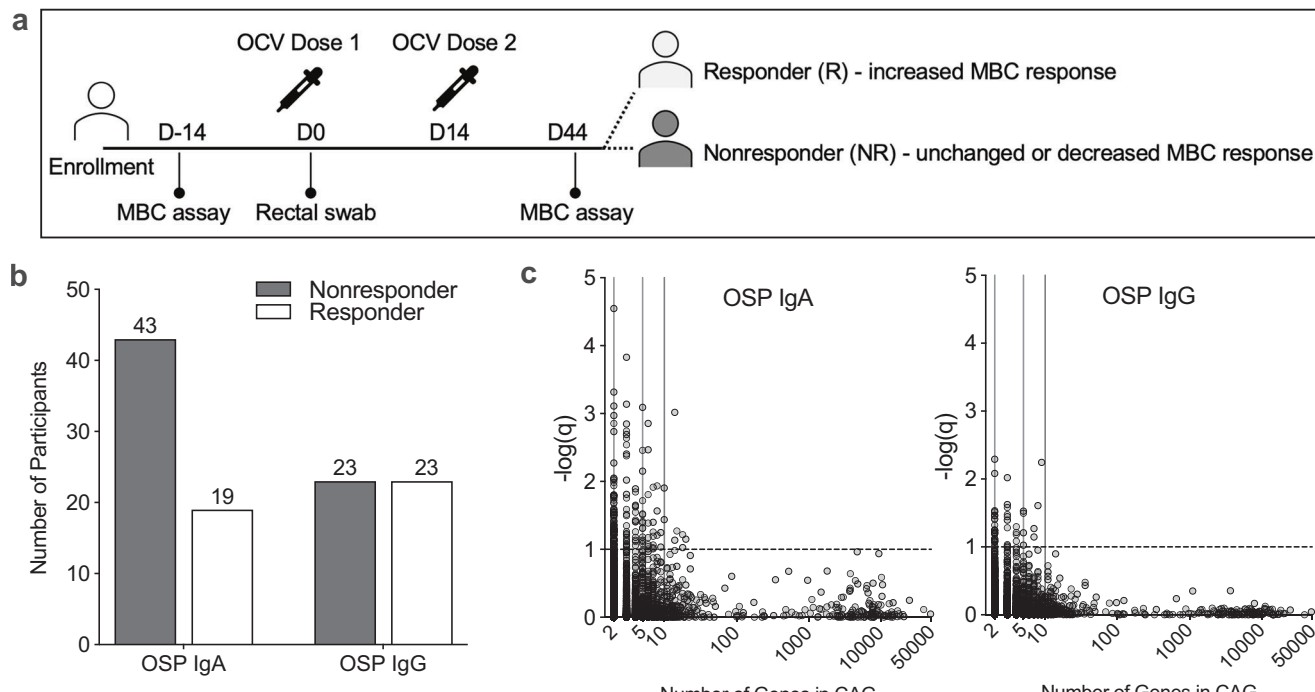

**Fig. 1 | Co-abundant groups of genes in vaccine participant stool are associated with protective memory B cell responses to OCV.** (**a**) Schematic of study design, OCV dosing, and sample collection timepoints, adapted from parent study[27]. Responders were defined as an OCV recipient who developed an increase in MBC response between pre- and post-vaccination timepoints, and nonresponders were defined as participants with unchanged or decreased MBC responses between timepoints. (**b**) Counts of vaccine responder and nonresponder participants by OSP-associated MBC responses. (**c**) CAGs are shown by size distribution and association with MBC response measures. CAGs are plotted as open circles based on the number of genes in a CAG along the *x*-axis and the -log of FDR-adjusted q-value along the *y*-axis. The dotted line corresponds to q = 0.1 and represents the pre-determined threshold for CAG significance. Source data are provided as a Source Data file.

years (11/22, 50% female), 22 participants >5-17 years (12/22, 54% female), and 45 participants ≥18-46 years (32/45, 71% female). Immune responses after OCV were consistent with prior studies of whole-cell killed vaccine responses from populations living in cholera-endemic areas[12,25,29,35]. As described in the parent study[27], vibriocidal responses to *V. cholerae* O1 Ogawa and Inaba serotypes increased after vaccination as expected, and only some participants had an increase in MBC response to vaccination, as observed in prior studies[26,36]. Among participants with MBC increases after vaccination, the mean increases in CT- and OSP-specific IgG-producing MBCs before and after vaccination were significantly increased, while CT- or OSP-specific MBCs producing IgA were not, as previously observed[27]. We found that OCV recipients were more likely to develop IgG-directed OSP-specific MBC responses than IgA OSP-specific responses (Fig. 1b). Among the immunologic parameters measured in this study, we focused primarily on *V. cholerae* OSP-specific MBC results because the presence of detectable OSP-specific MBCs is associated with long-term protection from infection, and a potential mechanism for OSP-specific antibody protective immunity has recently been established[9,17,18]. To assess microbiome differences among vaccine recipients, participants were grouped according to the development of an MBC response between pre- and post-vaccination timepoints for each (CT or OSP) IgA and IgG response. Individuals with increased MBC counts following vaccination were defined as vaccine responders (R), and participants who had unchanged or decreased MBC measures after vaccination were defined as vaccine nonresponders (NR).

We compared microbial taxonomic diversity measures by age and sex and found no significant differences by age (Supplementary Fig 1a-b). Shannon diversity (a measure of both species diversity and abundance) demonstrated a difference in microbial diversity by sex (Supplementary Fig 1c-e), driven by fewer *Lactobacillales, Veillonellaceae*, and *Eubacteriaceae*, and more *Erysipelotrichaceae, Ruminococcaceae*, and *Lachnospiraceae* in women, consistent with microbiome-specific sex differences noted in other populations[37]. No significant differences were found by Shannon Index or by Bray-Curtis community structure between vaccine responders and nonresponders for the four antigen-specific MBC measures (IgG- and IgA-producing OSP-specific or CT-specific MBCs, Supplementary Fig 2). A rarefaction curve, which shows the number of species observed in the study based on the number of sequences sampled, demonstrated that our sampling method captured the majority of the bacterial diversity in our study population (Supplementary Fig 3).

## Co-abundant gene (CAG) groups differentiate the microbiota of OCV responders from nonresponders

To address the high dimensionality of the metagenomic data and examine detailed features of the microbiota, we used gene groups as the fundamental unit of analysis. CAGs are likely to be encoded on a shared genetic element (e.g., chromosome or plasmid) across the organisms present in the cohort, and grouping genes together in this way has been successfully used to infer the relationship between specific genes and biological outcomes[38–40]. We identified CAGs that were associated with MBC responses and predetermined a significance threshold of a False-Discovery Rate (FDR)-adjusted q-value of ≤0.1, based on prior studies associating metagenomic data with biological responses (Fig. 1c)[41]. We calculated significant CAGs for each vaccine response measure separately (IgA OSP, IgA CT, IgG OSP, and IgG CT), and identified CAGs associated with an MBC response or lack of MBC response in each group. A total of 4147 CAGs were identified, and 323 CAGs containing ≥2 genes met our significance threshold across the four vaccine response measures (Supplementary Table 1). The greatest number of significant CAGs that mapped to bacterial genomes were associated with IgA OSP-specific MBC responses, and most of these CAGs were associated with a higher immune response (Supplementary Table 1). CAGs mapped to assembled bacterial genomes demonstrated

that many CAGs aligned to multiple strains (Supplementary Fig 4). Many genes composing significant CAGs have unknown functions or hypothetical proteins (based on the results of the eggNOG functional annotation and NCBI nr/blastp database (Supplementary Data 1)). Because all human sequences were removed from metagenomic data prior to CAG construction, this represents an enormous under-explored and unidentified microbial diversity of possible clinical significance that may be bacterial, fungal, or viral in origin. Among annotated genes in significant CAGs, most encoded nonspecific cellular functions, were related to amino acid metabolism, or were phage-associated.

## CAG group association with OCV responses was strain-specific

Because CAGs mapped to multiple reference strains, a priority score was assigned to each bacterial strain detected from participants in this study (see Methods). This process resulted in the identification of top bacterial strains associated with each MBC response, and many were from diverse taxonomic groups (Fig. 2a-b, Supplementary Fig 5a-b, Supplementary Table 2, list of strains with priority scores, Supplementary Data 2-5). Typically, only a small fraction of strains within a given taxonomic group had a high priority score, indicating that the association between MBC responses and gut microbes may be strain specific (Fig. 2c, Supplementary Fig. 5c). Among 412 total bacterial genera detected in the stool of the study population, only 34 contained five or more strains that mapped to a significant CAG (Fig. 2c, Supplementary Fig. 5c). To demonstrate the relationship between high-priority and unscored strains within one species, we identified *Bacteroides xylanisolvens* strains that were detected in the stool of our study population. To put these strains in the context of the observed diversity of this species, we next created a phylogenetic tree using NCBI genomes of all existing *B. xylanisolvens* strains (Supplementary Fig. 6). The four most highly ranked *B. xylanisolvens* strains identified by our analysis were not consistently clustered within the overall population of known *B. xylanisolvens* (Fig. 2d). This suggests that even closely related strains differ enough in gene content to be associated with different biological effects, and potential biological effects are not restricted to a single monophyletic lineage or clade. One exception to this finding is that most strains of *Clostridioides difficile* detected in our study population were highly positively associated with this MBC response (Fig. 2c). Among the 1,080 strains identified in study participant stool that positively correlated with OSP IgA MBC responses, *Bacteroides* species (n = 31) and *Prevotella copri* (n = 53) were disproportionately represented among the highest priority scored strains (Fig. 2b, Supplementary Data 2). Strains positively associated with OSP IgG responses were taxonomically varied and included members of *Bifidobacterium* and *Ruminococcus/Blautia* genera and several Enterobacteriaceae (including *Escherichia coli, Enterobacter*, and *Klebsiella* species) (Fig. 2a, Supplementary Data 3). Our results indicate that associations with biological outcomes were strain-specific, and thus, strain-specific methods of identifying these relationships are needed to select appropriate strains for experiments to test and validate a correlative host-microbe interaction. These findings support the concept that taxonomic groupings alone are unreliable markers for specific genes or associations with biological effects.

## The microbiota of vaccine responders was characterized by bacterial species with sphingolipid-producing potential

The identification of bacterial strains associated with OSP-specific IgA-producing MBC responses featured genera with the ability to produce sphingolipids, such as *Bacteroides* and *Prevotella* (Fig. 2b-d). We were intrigued by this because animal studies have demonstrated that *Bacteroides*-derived sphingolipids reduce intestinal mucosal inflammation[42,43]. This was also a notable finding because bacteria rarely produce their own sphingolipids, and these bacterial-derived

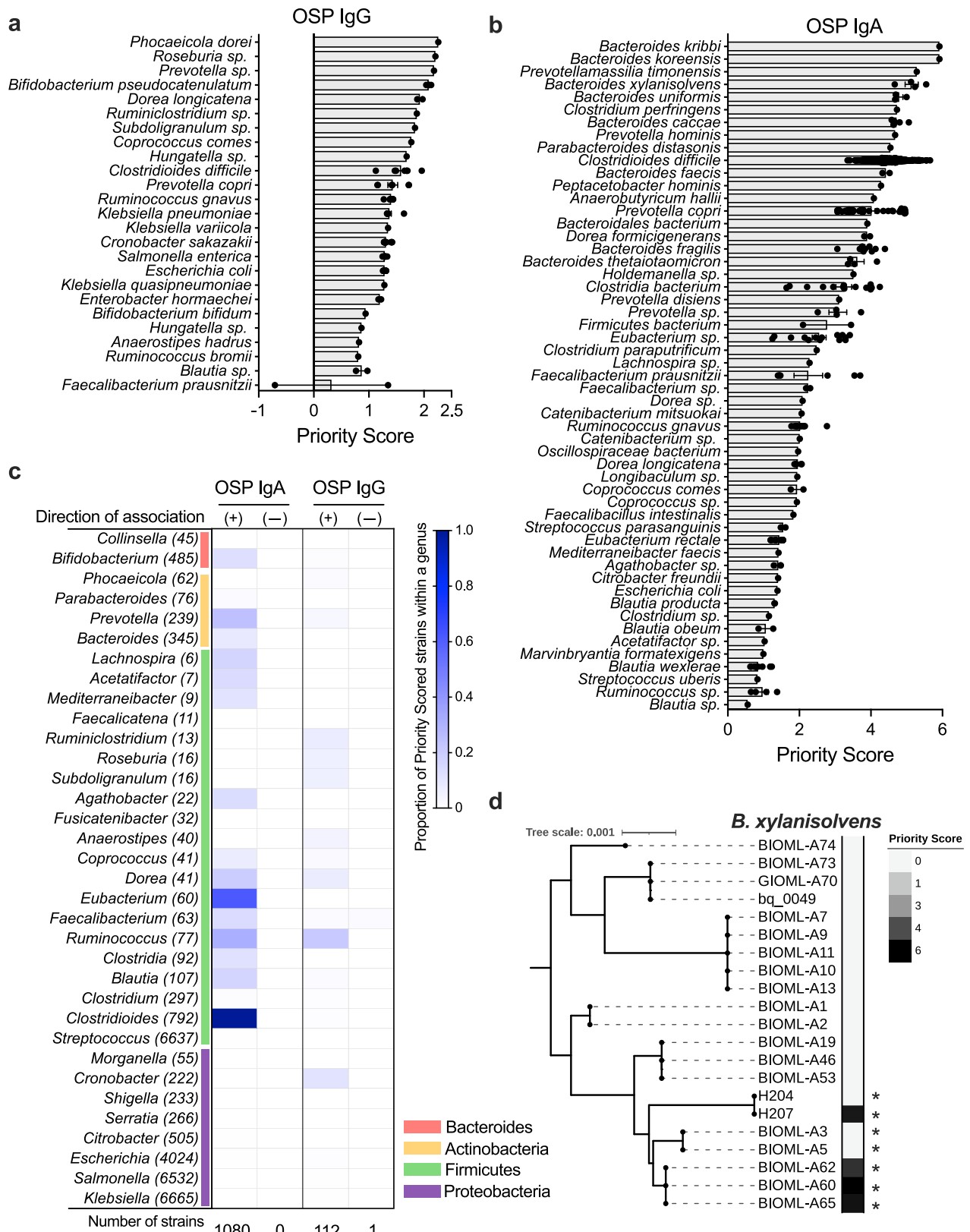

lipids are unique due to their odd-chain structures, in contrast to even-chain sphingolipids made by eukaryotes[44–46]. Because the metabolic capacity for sphingolipid production can vary widely across members of *Bacteroides*[47], we used the strain-level information identified by the CAG analysis to measure the relative abundance of organisms encoding the serine palmitoyltransferase (*spt*) gene, the bacterial enzyme

required for de novo assembly of microbe-derived sphingolipids[43]. We found that vaccine responder stool had a higher abundance of *Bacteroides spt* compared to nonresponder stool (Fig. 3a). For these reasons, we investigated the relationship between vaccine response and *Bacteroides*-derived lipids using a targeted, quantitative mass spectrometry-based lipidomic analysis of study participant stool.

**Fig. 2 | CAG groups associated with OSP-specific memory B cell responses are strain-specific.** Whole genome sequencing was performed on fecal samples from our cohort (n = 89) and analyzed using a method CAGs as described in the methods. CAGs were mapped onto all complete NCBI bacterial genomes as described in the methods, and species represented by significant CAGs with a priority score greater than 0.5 are shown for (**a**) OSP IgG and (**b**) OSP IgA MBC responses. Priority scores quantify the association between these strains and their association with a specific MBC response (see Methods). Each dot indicates the priority score of a specific strain within the species in the row. Individual strain priority scores are listed in Supplementary Data 2-5. Bar represents the mean with SEM of priority scored strains. (**c**) Proportion of strains found in our study population within each genus that were associated with a specific MBC response with positive (+) and negative (-) associations. Genera that contained 5 or more strains with significant CAGs were plotted with shading representing the percentage of total strains that contained significant CAGs. Parentheses indicate the total number of distinct strains in each genus found in our study population. Phylum identifications are shown in the colored vertical bar. For example, approximately 15% of the 485 *Bifidobacterium* strains detected in the study population were assigned a positive priority score associated with OSP IgA responses. (**d**) Priority scores differ widely among closely related strains within one species. Phylogenetic tree of *Bacteroides xylanisolvens* strains found in our study population and their priority scores representing their association with the OSP IgA MBC response. Tree made using NCBI published genomes of *Bacteroides xylanisolvens*. Asterisks (*) indicate the presence of serine palmitoyltransferase (*spt*) orthologs. Spt is the enzyme required for bacteria to de novo synthesize sphingolipids. Source data are provided as a Source Data file.

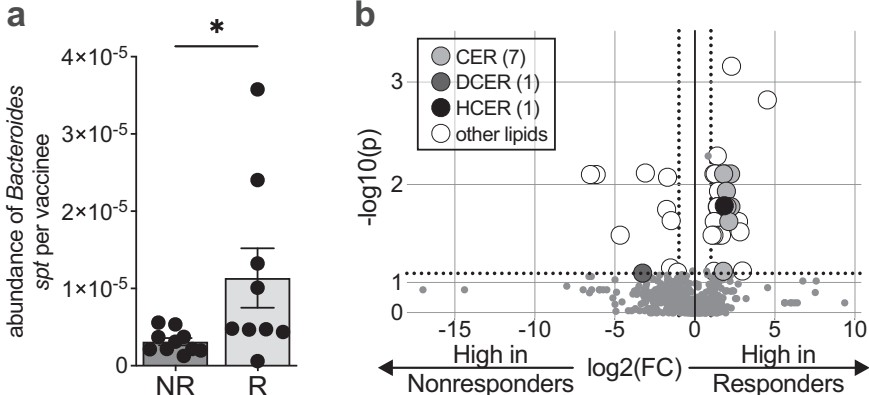

**Fig. 3 | Bacteroides *serine palmitoyltransferase* gene (*spt*) presence in stool is associated with a protective memory B cell response to OCV.** (**a**) Relative abundance of *Bacteroides spt* among vaccine responders (R) who had both an IgA- and IgG-specific OSP response to OCV and nonresponders (NR) who lacked both of these OSP responses. Each dot represents one participant. N = 10 nonresponders and n = 9 responders were included in our analysis of *spt*. Bars represent the mean with SEM. A two-sided Mann–Whitney test was performed. *, *P* = 0.0279. (**b**) Targeted quantitative mass spectrometry-based lipidomics of fecal samples from vaccine responders and nonresponders at the time OCV was administered. The quantities of the most abundant lipid species detected in study participant stool are listed in Supplementary Table 3. N = 7 nonresponders and n = 9 responders fecal samples were used for lipidomics. *P*-values from two-sided multiple Mann–Whitney tests are displayed on the *y*-axis with -log10 transformation, and the *x*-axis is fold change with log2 transformation. Dotted lines represent fold change of 1 or −1, and *P*-value of 0.05 along the x- and y-axis, respectively. CER: ceramides, DCER: dihydroceramides, HCER: hexosylceramides. Other lipids include sphingomyelin, cholesterol esters, diacylglycerols, free fatty acids, and triacylglycerols. Source data are provided as a Source Data file.

Several sphingolipid derivatives, including ceramides, were more abundant among vaccine responders (Fig. 3b and Supplementary Table 3). Ceramides are composed of a sphingoid base and a fatty acid, and bacterial sphingolipids are derived from this ceramide backbone[48]. Thus, ceramides are a structural backbone of all sphingolipids, and ceramide detections approximate sphingolipid content. There were also specific species of non-sphingolipid lipids that were more abundant in vaccine responders, including diacylglycerol, cholesterol esters, and free fatty acids, while triacylglycerols were found to be more abundant in vaccine nonresponders (Supplementary Table 3). While this stool lipidomics analysis does not fully distinguish between host- or bacterial-derived lipids, we were interested in differences in lipid content between OCV responders and nonresponders because in our prior study, fecal extracts from these two groups generated different innate immune responses in our model of OCV response[25], and our metagenomic analysis above suggested that bacterial-derived sphingolipids could be contributing to these findings.

### *Bacteroides*-produced sphingolipids blunted inflammatory cytokine production in human macrophages

Since innate immune responses impact the development of subsequent adaptive immune responses, including the differentiation and proliferation of MBCs, we next prepared to test whether *Bacteroides* sphingolipids impacted mucosal innate immune responses in our established human macrophage model of OCV response[25,49]. We first selected a *spt*-positive *Bacteroides xylanisolvens* (Bx) strain from our catalog of isolates derived from the stool of the study population and used this for additional experiments. We next generated lysates from Bx culture grown with or without the presence of myriocin, an *spt* inhibitor[42]. We then fractionated the Bx lysates to evaluate if lipid or non-lipid components of the lysate impacted immune responses in our model. Fractionation using the Bligh–Dyer method yielded culture metabolites, non-soluble cell membrane, and lipid components[50]. To ensure that myriocin had reduced sphingolipid production in the myriocin-treated Bx lipid extracts, we next measured lipid composition and abundance in the Bx lipid extracts using hydrophilic interaction liquid chromatography (HILIC) followed by ion mobility-mass spectrometry[51]. Bx lipid extracts exposed to myriocin had reduced ceramide abundance when compared to Bx lipid extracts not exposed to myriocin (Supplementary Fig. 7). We also sought to identify specific ceramides made by Bx and compared these to lipids found in vaccine responders' and nonresponders' stool. Lipid identification from stool has significant limitations, especially for full resolution of branching and structural patterns, and this is further complicated by incomplete knowledge about sphingolipids made by bacteria. Thus, full resolution of lipid structures from measurements in stool was not possible. However, we identified two bacterial-derived sphingolipids detected with high confidence, both in stool and in the Bx sphingolipid extract,

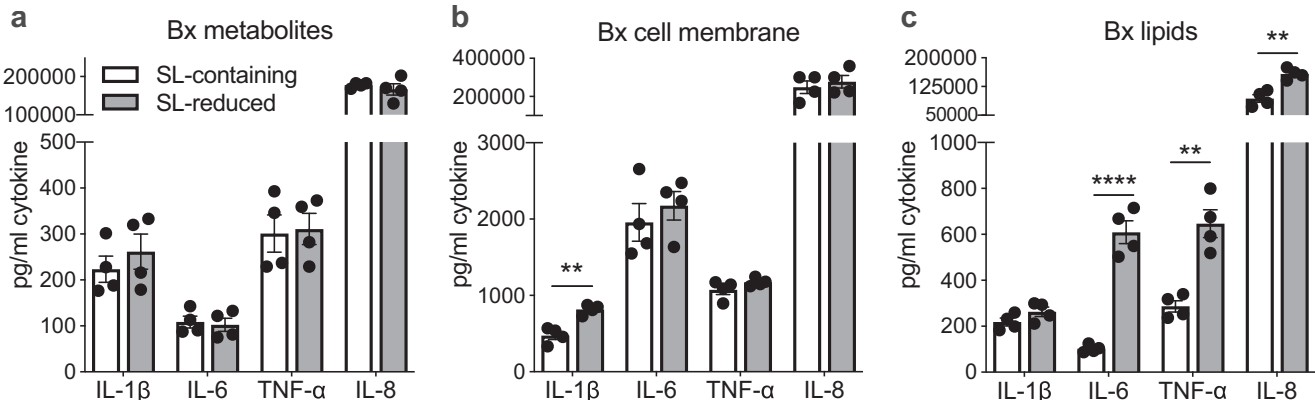

**Fig. 4 | Exposure to *B. xylanisolvens*-derived sphingolipids reduces immune activation of human THP–1 macrophages.** A *B. xylanisolvens* (Bx) isolate from a study participant's stool was cultured with or without myriocin (SL-reduced and SL-containing, respectively), and the bacterial lysate was fractionated using Bligh–Dyer lipid extraction. Cytokine responses were measured in the supernatant of THP-1-derived macrophages treated for 18 h with *B. xylanisolvens* (**a**) metabolites, (**b**) non-soluble cell membrane components, and (**c**) lipids. ELISAs were used to measure cytokine response. N = 4 technical replicates of THP-1-derived macrophages for each analysis. Statistical analysis was performed using two-tailed multiple unpaired *t*-tests with the FDR two-stage set-up method of Benjamini, Krieger, and Yekutieli. (**b**) IL-1$\beta$, *P* = 0.001544, q = 0.004679; (**c**) IL-6, *P* = 0.000056, q = 0.000056; TNF-$\alpha$, *P* = 0.001480, q = 0.000626; IL-8, *P* = 0.001860, q = 0.000626. Bars indicate mean with SEM. Source data are provided as a Source Data file.

CER(16:0) and CER(18:1). Both of these ceramides were more abundant in vaccine responders and more abundant in *B. xylanisolvens* lipid extracts untreated with myriocin (Supplementary Table 3 and 4).

After confirming that Bx lipid fractions treated with myriocin had reduced sphingolipids (Supplementary Fig. 7), we next examined the effect of sphingolipid-reduced (SL-reduced) Bx lysate and found significantly less interleukin-6 (IL-6) production compared to the untreated, sphingolipid-containing (SL-containing) lysate when applied to THP-1 macrophages (Supplementary Fig. 8). We then tested the fractionated Bx. Cytokine responses in our human macrophage model stimulated with metabolite and non-soluble fractions of Bx culture did not differ between SL-reduced and SL-containing fractions (Fig. 4a-b). We then tested the SL-reduced and SL-containing lipid fractions, and we identified that IL-6, tumor necrosis factor-$\alpha$ (TNF-$\alpha$), and IL-8 were decreased in the presence of Bx sphingolipids (Fig. 4c). The production of IL-6 by cells exposed to Bx SL-reduced lipids was nearly 6-fold compared to cells exposed to the SL-containing lipid fractions. We also evaluated cell viability in the presence of lipid treatment and found that lipid extracts did not alter cell viability (Supplementary Fig. 9).

### *Bacteroides*-produced sphingolipids increased innate immune responses to *V. cholerae* antigens

We observed an increase in innate cytokine responses when Bx SL-reduced lipid extracts were applied to human macrophages, and this was consistent with our prior findings that fecal extracts from vaccine nonresponders increased IL-6 production in this model[25]. Because this response was triggered by vaccine nonresponders only, we hypothesized that an increase in resting innate immune activation could dampen the immune response to vaccine antigens. To test this, we pretreated THP-1-derived human macrophages with SL-reduced and SL-containing lipid fractions and then exposed them to heat-killed *V. cholerae* strain JBK70, an A-B toxin negative vaccine strain, as a proxy for OCV, as in prior studies[49]. Macrophages that were pretreated with SL-containing lipid fractions and then stimulated with JBK70 were found to produce increased IL-6 and TNF-$\alpha$ (Fig. 5a). In contrast, macrophages pretreated with SL-reduced lipid fractions had lower cytokine responses to JBK70. To minimize the possibility that this effect was caused by non-SL lipid species, we next depleted phospholipids from the lipid extracts using a mild alkaline hydrolysis treatment[42]. This change potentiated the macrophage response to

sphingolipids, with further increases in the magnitude of IL-1$\beta$, IL-6, IL-12p40, and TNF-$\alpha$ production in response to JBK70 stimulation (Supplementary Fig. 10). To test for this effect in another *spt*-positive *Bacteroides* isolate from our study population, we selected a strain of *Bacteroides koreensis*, a species identified as highly associated with OSP-specific MBC OCV responses (Fig. 2b). Our findings were replicated using this strain (Supplementary Fig. 11).

To further query the effect of bacterial sphingolipids on human macrophages, we next performed a gene expression analysis on RNA extracted from THP-1 macrophages preconditioned with Bx SL-containing or SL-reduced lipid fractions with and without exposure to heat-killed JBK70 (Fig. 5b). Using a Gene Set Enrichment Analysis[52], we found that preconditioning macrophages with SL-reduced Bx lipids prior to JBK70 exposure increased activation of several inflammatory pathways compared to cells preconditioned with SL-containing lipids (Supplementary Fig. 12a). Compared to the SL-containing fraction, SL-reduced fraction preconditioning triggered pathways associated with interferon-$\alpha$, interferon-$\gamma$, and TNF-$\alpha$ cytokine responses, and IL-6−Janus kinase (JAK)−signal transducer and activator of transcription (STAT3) pathways. After preconditioning and subsequent exposure to heat-killed JBK70, TNF-$\alpha$ signaling via the nuclear factor kappa-light-chain-enhancer of B cells (NF-κB) pathways was upregulated independent of SL exposure (Supplementary Fig. 12b, c, individual genes listed in Supplementary Table 5).

To identify immune activation pathways that distinguished SL-reduced from SL-containing lipid stimulation of human macrophages after JBK70 stimulation, we repeated our analysis to exclude gene expression specific to heat-killed JBK70 that was independent of bacterial sphingolipid exposure. Here we found that human macrophage gene expression after SL-containing lipid treatment and JBK70 stimulation was enriched in TNF-$\alpha$ signaling via NF-κB, IL-6−JAK−STAT3 signaling, inflammatory response, epithelial/mesenchymal transition, and complement pathways, compared to the macrophage response after treatment with SL-reduced lipids (Fig. 5c). The 19 genes differentially associated with enriched pathways are shown in Fig. 5d and Supplementary Fig. 13. Several of the genes were associated with multiple innate inflammatory pathways, including inhibin A (INHBA), C-X-C motif chemokine ligand 7 (CXCL7), and CXCL1.

Lastly, to validate our findings in THP-1 macrophages using additional immune cell types, we tested the effect of OCV antigens with and without sphingolipid exposure in human peripheral blood

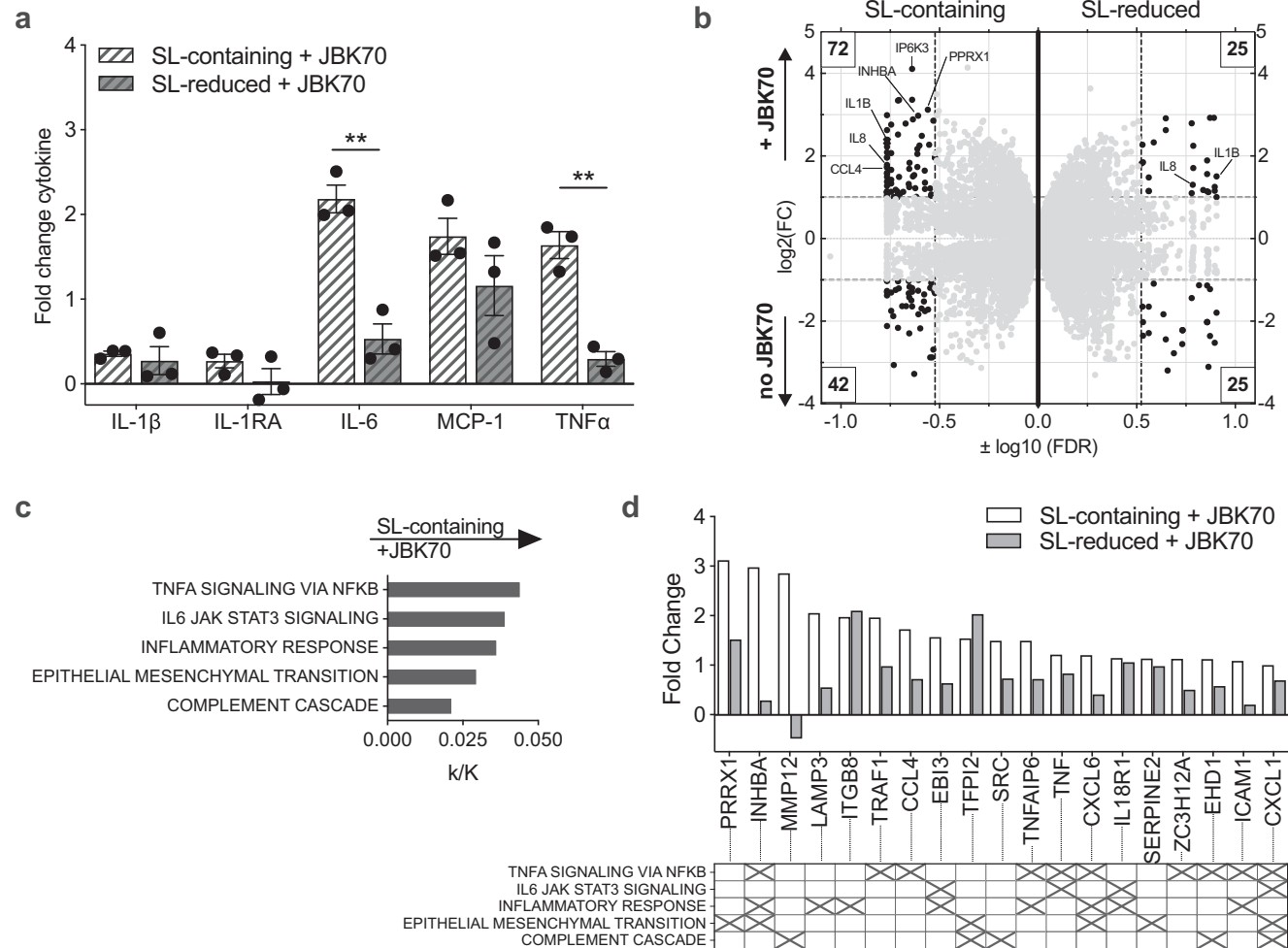

**Fig. 5 | *B. xylanisolvens*-derived sphingolipids increase inflammatory cytokine responses from THP-1 human macrophages after stimulation with a *V. cholerae* vaccine strain. a** Cytokine responses are shown as fold change measured in supernatant from THP-1-derived macrophages following preconditioning with Bx lipid extracts and treated with heat-killed JBK70. N = 3 technical replicates of THP-1-derived macrophages. Bars indicating the mean with SEM are shown. Two-tailed unpaired *t*-test comparing fold change; IL-6, *p* = 0.002348; TNF-*α*, *p* = 0.001730. **b** Gene expression from THP-1-derived macrophages was measured by RNA sequencing following pre-conditioning with Bx SL-containing or SL-reduced lipid fractions with or without exposure to HK JBK70. *x*-axis is log10(FDR), and *y*-axis is log2(fold change). Each gene is represented as one dot and is shown twice in this figure, once in the SL-containing (left side), which was preconditioned with sphingolipids, and once in the SL-reduced (right side), which was preconditioned without sphingolipids present. Numbers boxed in the corners indicate the number of genes with FDR < 0.3 and FC > 2. **c** Pathways enriched based on macrophage gene expression after treatment with SL-containing lipid fractions and JBK70 stimulation, compared to macrophages treated with SL-reduced lipid fractions, FDR values < 0.05. The *x*-axis label is proportion enriched (k/K), represented as the ratio of significant genes in that pathway over total genes in the specified pathway, calculated using BIGprofiler through clusterProfile. **d** Fold change of genes identified in pathways enriched following SL-containing lipid fraction exposure and JBK70 treatment in THP-1 macrophages. The matrix below the graph indicates gene membership in each pathway based on hallmark gene sets in the Molecular Signatures Database. Source data are provided as a Source Data file.

mononuclear cells (PBMCs). We preconditioned PBMCs with *B. xylanisolvens* sphingolipid fractions and replicated our findings of increased IL-6 after JBK70 stimulation, as well as TNF-*α* in some donors (Supplementary Fig. 14). These data in combination support our hypothesis that the correlations we observed between sphingolipid-producing bacteria and vaccine response in humans may be partly attributable to sphingolipid-driven modulation of the innate immune responses to OCV antigen.

## Discussion

Oral cholera vaccines are an essential tool for reducing the global impact of cholera. Yet, the protection they provide is limited and differs between individuals for unclear reasons. Improving the duration and degree of protection afforded by currently available OCVs would be a critical advance in cholera control. We previously found that the

microbiome at the time of OCV administration was associated with the long-term MBC responses that are thought to mediate protection from cholera[25]. To explore the relationship between the gut microbiota and host immune responses to OCV, we used whole genome sequencing (WGS) of the stool microbiota and identified the relationship between CAGs with our biological outcome of interest. We then applied an original priority scoring method that allows highly ranked CAGs to be mapped to specific strains, controlling for CAG size, gene abundance, and other sequencing factors. We detected an association between commensal gut microbes that produce bacterial sphingolipids in stool from persons who developed protective immune responses after OCV, and in vitro experiments supported this finding and demonstrated greater innate immune responses to heat-killed JBK70, a *V. cholerae* vaccine strain. Taken together, these results suggest that gut microbial sphingolipids could enhance innate immune responses to oral cholera

vaccines. Increasing the immunogenicity of existing cholera vaccine candidates that currently make up the World Health Organization vaccine stockpile would be a critical advance in the context of current vaccine shortages and increasing cholera outbreaks worldwide.

Previous studies of the relationship between the gut microbiota and responses to oral vaccination used sequencing methods that did not distinguish metagenomic content well enough to identify strain-level associations with vaccine response[30,31,53,54]. Here, we describe a quantitative method of analyzing metagenomic sequence data that enables the identification of bacterial strains associated with a biological outcome without dependence on a reference genome database or taxonomy for results interpretation. Because interpreting the biological significance of de novo assembled gene content has serious limitations, we followed the CAG-level outcome association with exhaustive alignment of gene content against a collection of reference genomes. In this manner, we identified patterns of association tied to gene content that were orthogonal to a strict taxonomic hierarchy of microbial diversity. Supporting the value of our taxonomy-independent approach in this analysis, we observed no differences in alpha or beta diversity between vaccine responders and non-responders for any of the vaccine response measures, and we found that strain-specific associations were not generalizable by genera or species.

To experimentally test our metagenomic study findings, we first examined the relationship between vaccine response and the abundance of bacterial genes in stool that indicated the presence of sphingolipid-producing microbes. Based on these results, we experimentally tested relevant bacterial strains that were associated with OSP-specific MBC responses to vaccination. Sphingolipid-producing microbes are generally represented in the human gut microbiota by two prominent commensal genera, *Prevotella* and *Bacteroides*[48]. Microbe-derived sphingolipids from *Bacteroides* were previously observed to impact mucosal immune responses; for instance, Brown and colleagues found that the lack of *B. thetaiotaomicron* sphingolipids in mono-colonized mice resulted in increased IL-6 and MCP-1 produced ex vivo from epithelial cells[43]. We identified a similar pattern of increased IL-6 production in our model of OCV responses using human macrophages exposed to Bx sphingolipid-reduced lipid fractions. Based on the stimulatory response to Bx lipids that we observed in human macrophages, we were surprised to observe that exposure to sphingolipid-reduced Bx lipid fractions resulted in lower innate responses to a heat-killed *V. cholerae* strain used as a proxy for vaccine antigen. This inverse relationship, wherein higher baseline innate immune activation is correlated with a dampened immune response to vaccine antigen, has been previously described[55–57]. For example, higher systemic innate immune responses at the time of Hepatitis B virus (HBV) vaccine administration correlated with lower peripheral immune responses in humans[58,59]. HBV vaccine nonresponders were found to have much higher pre-vaccination gene expression of innate immune effectors and proteins, including immune responses to LPS and defensins that activate NF-κB signaling pathways[60]. This effect has also been observed in human yellow fever vaccination, in which systemic measures of lymphocyte activation and high levels of proinflammatory monocytes at baseline negatively correlated with neutralizing antibody titers post-vaccination[61]. This relationship has not been previously observed in oral vaccination, and warrants further exploration as a potential tool in improving responsiveness to OCVs and other oral vaccines. Although more testing is needed, the approach of using commensal gut microbes to condition or prime the mucosal surface for optimal responsiveness to vaccine antigens is novel and distinct from the use of traditional adjuvants that act by causing increased inflammation at the site of antigen delivery.

We measured the impact of microbial sphingolipids in our model of human innate immune responses to OCV because these responses support the differentiation and maintenance of antigen-specific B cells that mediate long-term protection from cholera. Sphingolipid-producing microbes were prominent among microbes associated with increased OCV responses in our analysis, and the plausibility of this relationship is supported by evidence from animal studies in which the presence of microbial sphingolipids was associated with more robust adaptive mucosal immune responses[62,63]. When we exposed human macrophages to a heat-killed OCV strain after pre-conditioning the cells with Bx sphingolipid-containing lipid fractions, we observed activation of several innate pathways characterized by genes typically expressed in bacterial infection and bacterial clearance[64–71], and this was accompanied by expression of genes involved in inflammatory cell recruitment including leukocytes (via ICAM1), neutrophils (via CXCL1 and CXCL6), and dendritic cells and T cells (via C-C motif chemokine ligand 4 (CCL4))[64,72–74]. CCL4 is associated with the migration of CD4 + T cells and enhanced mucosal secretion of antigen-specific IgAs[75–77]. This pattern of innate immune activation was significantly reduced when these lipids were absent at the time of OCV introduction, and downstream studies of these effects on other cell types are needed, in addition to the identification of the microbe-derived lipid species that produce this response.

Several CAG groups that were highly associated with immune responses to OCVs in our study mapped to pathogens, such as strains in the *Shigella* genus and strains of *Salmonella enterica*. Persons with diarrhea were excluded from this study, and these pathogens might have been detected in the stool due to subclinical or recent infection, or could be a marker of exposure to contaminated food and water, as has been described in other study participants living in areas lacking modern sanitation[78–82]. Notably, we found that many strains of *Clostridioides difficile*, which can be a colonizing microbe in this population, were highly associated with IgA-specific OSP responses, but not other immune responses[78]. Two *C. difficile* peptides (the fragment of the receptor-binding domain of toxin A TxA[C314] and a fragment of the 36 kDa surface-layer protein [SLP-36kDa] from strain C253) are known to have immunoadjuvant properties; investigators previously noted an increase in mucosal IgA and systemic IgG targeting antigen when these peptides were co-administered to mucosal surfaces in mice[83]. This mechanism is a potential explanation for the strong correlation observed in our study between *C. difficile* and OSP IgA responses, and the specific pathways activated in response to OCV antigen will require further study. As expected, we found differences in the gut microbiota between males and females in this study; this was independent of overall immune responses to vaccination, and CAG distribution did not differ between sexes. We also did not collect data on antibiotic use or helminth infection, which are both widespread in this population, and could potentially impact immune responses[84,85]. We were not able to control for the presence or absence of helminths in the study participant microbiota using sequencing data due to the limitations of metagenomic analysis, including a relatively low abundance of eukaryotic reads compared to bacterial reads and a lack of accurate reference helminth genomes[86,87]. While we observed an increase in ceramides in the human fecal lipidomics among vaccine responders, other stool lipids were also increased in OCV responders, including diacylglycerol, cholesterol esters, and free fatty acids, many of which are known to be host-derived lipids[48] and may or may not be related to the microbiota. Triacylglycerols were more abundant in OCV nonresponders, and this class of fecal lipids are a marker of malabsorption[88], raising the possibility that absorptive status could impact OCV response. It is possible that host-derived sphingolipids in stool that are unrelated to the microbiota may also be responsible for the findings we observed in human stool. Further investigation of these lipids could provide insights into host factors associated with vaccine response.

In conclusion, we used a reference-independent gene-level metagenomic analysis of the gut microbiota at the time of vaccination to pinpoint specific strains that may govern the relationship between

gut microbes and OCV responsiveness. Our findings strengthen the concept that individual differences in the microbiota drive variation in OCV response. Modification of baseline innate immune activation mediated by gut microbes could be a new target for improving OCV responses, such as through the use of pre- or probiotic products to be co-administered with OCVs.

## Methods

### Study design, subject enrollment, and sample collection

Adult and child participants over 2 years of age and <60 years of age living in urban Dhaka, Bangladesh, were enrolled through the International Centre for Diarrheal Disease Research, Bangladesh (icddr,b) between Jan 1, 2017, and Dec 30, 2017, if they had not previously been vaccinated with an OCV and had no major comorbid conditions[89]. After obtaining written informed consent from patients or guardians, study participants received two doses of WC-rCTB (Dukoral), per the package insert, on study Day 0 and Day 14. All timepoints are reported in reference to Day 0, the day of administration of the first dose. Blood specimens were collected from the participants on Day 14 (two weeks prior to the first dose of vaccination), Day 2, Day 7, and Day 44. Stool samples were collected on Day 0 on site at the icddr,b in cryovials and immediately frozen at −80 °C. Vibriocidal titer was assessed at each of these timepoints; OSP-specific and CT-specific MBC were measured at Day 14 and Day 44. Demographic data were collected at enrollment, and stool samples were collected at baseline prior to the first dose of WC-rCTB. Additional details and methods for vibriocidal titers, serum antibodies, and MBC measures are described in the parent study[27].

### Stool processing and DNA preparation

Stool samples were selected for sequencing if the participant had MBC ELISPOT results for both pre-and post-vaccination timepoints for at least two out of four MBC responses. Application of this criterion resulted in sequencing the stool from 90% of study participants. Stool was collected in cryovials and immediately frozen at −80 °C and shipped to US investigators. At the University of Washington, stool microbial DNA was extracted using the PowerSoil DNA isolation kit (Qiagen) with a modified protocol as previously described in ref. 78. Briefly, the stool was thawed on ice, and approximately 100 mg of stool was added to bead-beating tubes. The stool was treated with C1 solution, heated at 65 °C for 10 min, 95 °C for 3 min, and vortexed for 10 min. Samples were treated using C2–C5 solutions provided by the Powersoil kit. The resulting DNA was eluted in DNase- and RNase-free water; quality and quantity were measured using a NanoDrop ND-1000 (Thermo Scientific).

### Library preparation, sequencing, read processing, and CAG grouping

Library preparation was performed using uniquely dual-indexed libraries using the Nextera DNA Flex kit. Sequencing was performed on the NovaSeq 6000 PE150 platform using standard Illumina adapters targeting a minimum of 10 M reads per sample with 150 bp pair-end reads in one sequencing run. Fastq files were processed using *gene-shot*, a pipeline developed to streamline the analysis of gene-level metagenomic data and its association with experimental metadata[38]. Using default settings for this pipeline (v0.9), raw WGS data from each individual specimen was quality-filtered and de novo assembled using Megahit[90]; human reads were excluded using human reference genome GRCh37 hg19, and protein-coding sequences were predicted using Prodigal[91] and deduplicated across all specimens using linclust (at 90% amino acid similarity); coding sequences were annotated with taxonomic labels using DIAMOND (by alignment against NCBI RefSeq 01/22/2020) and with functional labels using eggNOG-mapper v2 (accessed 05/22/2024)[92] and NCBI blastp (accessed 05/22/2024) with database nr_clustered(experimental)); WGS reads from each specimen were aligned against the de novo generated gene catalog with

DIAMOND; coding sequences were grouped by co-abundance into CAGs[39]. The centroid was selected for each deduplicated gene using the mmseqs2 algorithm[93]. The depth of sequencing used for CAG generation was an average of the depth of sequencing across all genes in each CAG. The relationship between CAGs containing ≥2 genes was correlated with participant vaccine response measures using the corncob algorithm implementation of beta-binomial regression and accounting for multiple comparisons using the Benjamini-Hochberg false-discovery rate correction procedure[94,95]. CAGs containing only one gene were excluded to reduce the total number of tested hypotheses, considering the relative lack of biological interpretability for singleton coding sequences in the context of gene-level metagenomic analysis. Top CAGs were defined prior to the analysis as those with a $q$-value ≤ 0.1.

### Assessment of gut microbial communities

Reads from all participant samples were used to estimate the relative abundance of taxonomic groups using metaphlan2[96], which uses the alignment of raw reads against a set of previously identified marker genes that are conserved across taxonomic groupings of genomes. The output of metaphlan2 was merged across all samples, and non-bacterial taxa were removed. Alpha diversity measures by Shannon Index and beta diversity measures by Bray-Curtis Dissimilarity Index were calculated using divnet (v3.6) in R (v4.0.2)[97]. Principal coordinate of analysis (PCoA) plots to visualize these results were generated using the R package vegan. Rarefaction on species-level identification was calculated using mothur (v.1.44.1) command rarefaction.single() with 10 iterations/randomizations with sequence sampling frequency set at 100[98].

### CAG alignment to bacterial genomes

Top CAGs found to be significantly associated with vaccine response measures were aligned to bacterial genomes to identify the strains containing the most top CAGs. A strain is defined here as a single reference genome present in the RefSeq database. A total of 61,918 bacterial reference genomes were selected from the National Center for Biotechnology Information prokaryote genome database using DIAMOND v0.9.10 via the AMGMA (Annotation of Microbial Genomes By Microbiome Association) workflow [https://www.ncbi.nlm.nih.gov/genome/browse#!/prokaryotes/] and [https://github.com/fredhutch/amgma][39,99,100]. We used all bacterial sequences in the database apart from bacterial sequences with low assembly quality and completeness (chromosomes and scaffolds). We included all complete genomes and 85% of contig-level assemblies and excluded 15% of contig-level assemblies with the lowest levels of completeness (> 200 contigs). We expect that genomes with lower assembly quality will be more difficult to detect from metagenomic datasets using this or any other analytical approach. Because this database contains multiple representatives from many species and genera, it is expected that homologous copies of many CAGs may be found in multiple genomes. For that reason, alignments of a single CAG to multiple genomes were all retained for the analysis. Using the alignment of coding sequences from the metagenomic analysis against this set of partial and full genomes, a relative abundance measure was generated for each genome in each sample by summing the proportion of reads from a given sample that aligned to those genes found in each genome. Using that strain-level relative abundance measure, we applied the same statistical approach (beta-binomial modeling using the corncob algorithm) to estimate the coefficient of association with vaccine response measures on a per-genome basis. Our gene-based analysis described here is reference-free due to our method of CAGs generation and quantification across metagenomes as a function of vaccine response; the CAG alignment to bacterial genomes is based on the genomes contained in the RefSeq database.

## Development of a CAG-based strain identification method

Once CAGs were aligned to bacterial genomes[38,39], we next quantified the association between these strains and vaccine response measures. For this purpose, we created a priority score designed to (1) account for the strength of the association between top CAGs aligning to a genome and the vaccine response measure of interest, (2) preserve the directionality of the association between the vaccine response measure and top CAGs, (3) be independent of CAG size, in order to avoid bias toward large CAGs containing more genes that could potentially align, and (4) discriminate between strains within the same species.

For a given strain with $n$ number of significant CAGs aligning to strain genomes, this priority score was calculated as follows:

$$PS_{strainX} = \frac{\sum_{CAG1}^{CAGn}(Num\_Genes\_Align_{CAG1} \times \pm \sqrt{est\_coefficient_{CAG1}^2 + wald_{CAG1}^2})}{\sum_{CAG1}^{CAGn} Num\_Genes\_Align_{CAG1}}$$
$$\times \sqrt{mean\_est\_coefficient_{StrainX}^2 + mean\_wald_{StrainX}^2}$$

The estimated (est) coefficient reflects the abundance of a CAG in responders compared to nonresponders for each vaccine response measure, and the Wald score integrates the standard error, which reflects the uniformity of the association[39]. The $\pm$ in the first term corresponds to the sign of the CAG's estimated coefficient. The number of genes aligning from a CAG to the strain genome is represented in the estimated coefficient and Wald score, creating a weighted measure of CAG-strain association, which is summed in the first term's numerator for all top CAGs with genes aligning to the strain. By normalizing this weighted sum in the numerator of the first term by the unweighted number of genes aligning from top CAGs, we minimize bias based on CAG size. In the second term, the score is further weighted by integrating the number of genes aligned to reference strains without regard for CAG grouping, and thereby discriminates between strains that may have otherwise identical first terms due to the alignment of the same number of genes from the same top CAG(s). A + priority score indicates a positive association between the vaccine response measure and strain, and a - indicates an inverse relationship.

## *spt* gene quantification

*Bacteroides* serine palmitoyltransferase (*spt*) abundance was determined by querying BT_0870, a confirmed spt gene in *Bacteroides thetaiotaomicron*[43] on EggNog v5.0 (online)[92]. The orthologous group was identified as COG0156 (Amino acid transport and metabolism: 8-amino-7-oxononanoate synthase activity), and abundance was acquired using a Python script called geneshot_extract_gene_abund.py [https://gist.github.com/sminot/cebfbd84d57406b5b41b2eebffb1789f] on the geneshot results of the metagenomic data[38]. *Bacteroides fragilis* serine palmitoyltransferase (encoded by GenBank# EXZ60402.1) was queried among the *B. xylanisolvens* reference strains found in our metagenomic sequencing. *B. xylanisolvens* strains were subjected to NCBI BLASTP [http://blast.ncbi.nlm.nih.gov] using the default parameters. The top hits of each sample are listed with scores (bits) and E-values in Supplementary Table 6. Samples with scores > 500 and E-values less than E-50 were considered a hit, i.e., containing a close homology to the query.

## Fecal lipid quantification through lipidomics

Lipidomics was performed on human fecal samples using 50–100 mg of feces (Northwest Metabolomics Research Center at the University of Washington). Feces were heated at 70 °C for 8 h in a dry oven. Samples were then homogenized in a fixed volume of water, and a volume equivalent to 10 mg was used. Lipidomics was performed with the Sciex 5500+ with a SelexIon unit as described previously[101]. Data were plotted using log2 fold change of average concentrations of lipid species (μM) of OCV responders (n = 9) and nonresponders (n = 7).

Log10 (*P*-value), which was obtained using multiple Mann–Whitney tests.

## Bacterial strains, growth conditions, and identification

*Bacteroides xylanisolvens* was isolated from human fecal samples from the study participants[27] using brain-heart infusion (BHI) agar (BD) supplemented with 5% sheep's blood (Hemostat Labs) and Laked Brucella Blood agar with Kanamycin and Vancomycin (LKV) plates (Anaerobe Systems). *Bacteroides koreensis* was isolated using Gifu Anaerobic Media (HiMedia) supplemented with porcine hemin (10 mg/L; MP Biomedicals) and menadione (5 mg/L; MP Biomedicals) and LKV plates. Bacterial identity was confirmed using ~1000 bp Sanger sequencing of the 16S rRNA gene with >97% identity. Liquid cultures of *Bacteroides* strains were prepared in BHI broth (BD Diagnostic Systems) supplemented with 0.005 mg/mL porcine hemin, 0.5 g/L L-cysteine (Sigma), and 0.05 ng/mL Vitamin K1 (Sigma) and cultivated at 37 °C in a vinyl anaerobic chamber (Coy) with <50 ppm $O_2$. Glycerol stocks were prepared using BHI supplemented media, 20% glycerol, and stored in -80 °C ultra-low freezers. *Bacteroides* strains were maintained on Tryptic Soy Agar with 5% sheep's blood plates and grown in BHI supplemented media for 24–48 h. The *V. cholerae* strain JBK70 was gifted by Dr. Jim Kaper[49]. *V. cholerae* was maintained in lysogeny broth (LB) or LB agar at 37 °C in aerobic conditions.

## Mammalian tissue culture cell line and maintenance

THP-1 cells were purchased from and authenticated by ATCC (ATCC cat# TIB-202). PBMCs were purchased from Bloodworks Northwest. Four PBMC donors were acquired. All cells were maintained in RPMI-1640 media (containing L-glutamine and 25 mM HEPES) (Corning) supplemented with 10% fetal bovine serum (Fisher) and penicillin and streptomycin (Gibco) in 5% $CO_2$ and 37 °C conditions.

## Bligh–Dyer lipid extraction and mild alkaline hydrolysis

To inhibit *spt* and de novo sphingolipid production, 5 μM myriocin (Cayman Chemicals) was added to bacterial culture. Forty-eight-hour liquid bacterial cultures in BHI supplemented with or without myriocin with OD > 1.00 were used for lipid extraction. Bacterial cultures were normalized to an optical density (OD 600 nm) of 1–1.5 using sterile phosphate-buffered saline (PBS; Corning) with a total volume of 5 mL. Samples were then centrifuged at 4000×*g* for 20 min at 4 °C. The supernatant was removed, and the bacterial pellets were resuspended in sterile PBS and transferred to 1.5 mL microtubes and washed twice using PBS. Lipids were then extracted using the Bligh–Dyer method[50]. Briefly, bacterial pellets were then resuspended in 100 μL sterile water and treated with 400 μL chilled 1:2 parts chloroform:methanol (Fisher Chemical). Samples were vortexed for 5 min at max speed using a vortex microtube adapter. Samples were then treated with 100 μL chilled chloroform and 100 μL sterile molecular-grade water (Corning) and vortexed for 1 min. Samples were then centrifuged at 2000×*g* for 10 min. The bottom layer containing chloroform and lipids was collected into a new tube and air-dried in a sterile environment. Samples were stored at −80 °C until resuspended in 500 μL of cell culture media corresponding to the mammalian cells used for the assay. Reconstituted lipids were stored at −20 °C. Mild alkaline hydrolysis for the removal of phospholipids from lipid extracts was performed as previously described in ref. 42. Briefly, the lipid extracts before drying were treated with 0.02 N sodium hydroxide (NaOH; Acros Organics) for 30 min at 37 °C. Lipids were then dried, stored, and reconstituted as above.

## Lipidomics analysis of *Bacteroides xylanisolvens* cultures

*B. xylanisolvens* cultures were grown in technical triplicates (n = 3) for myriocin and no myriocin treatment. Bacterial cultures were pelleted by centrifugation, washed by resuspension and centrifugation in PBS, and total lipids were extracted using the Bligh–Dyer method[50]. Briefly,

extracts were dried in a vacuum concentrator, then reconstituted in 2:1 acetonitrile-methanol. Lipids were analyzed by HILIC coupled with ion mobility-mass spectrometry (IM-MS). Chromatographic separations were carried out with a Phenomenex Kinetex HILIC column (50 × 2.1 mm, 1.7 μm) on a Waters Acquity FTN UPLC (Waters Corp., Milford, MA) with 95% acetonitrile/5% water/5 mM ammonium acetate as mobile phase A and 50% acetonitrile/50% water/5 mM ammonium acetate as mobile phase B[51,102,103]. Collisional cross-section calibration and IM-MS analysis were conducted on a Waters Synapt XS HDMS (Waters Corp.) in the positive ionization mode using a series of phosphatidylcholines, covering the range of $m/z$ 454–980 and CCS 213–330 Å², as the calibrants and negative ionization mode using a series of phosphatidylethanolamines, covering the range of $m/z$ 410–786 and CCS 199–274 Å², as the calibrants[102–104]. Data alignment and peak detection were performed in Progenesis QI (Nonlinear Dynamics; Waters Corp.) with normalization to all compounds. Retention time calibration and lipid identification were calculated with the Python package LiPydomics (version 1.6.8) with default settings [https://github.com/dylanhross/lipydomics][105].

### THP-1 cell differentiation and treatment with bacterial lipids and JBK70

THP-1 cells were treated with 50 ng/mL PMA (phorbol 12-myristate 13-acetate) (Invitrogen) and plated into 24-well tissue-culture treated plates using $5 \times 10^5$ cells/well. Cells were incubated for 48 h prior to exposure to additional stimuli. Adherent cells were rinsed using sterile PBS, and the cell media was replaced. Bacterial lipids were applied to differentiated THP-1 cells in 1:100 concentrations for 24 h for pre-conditioning. Cells were rinsed with sterile PBS and then treated with heat-killed JBK70 (1:10) for a final dilution of 1:100. After 18 h, the supernatant of the cells was collected. For cell counts, adherent cells following lipid treatment were rinsed with sterile PBS and removed using Trypsin-EDTA (Corning). Cells were counted on a hemocytometer with Trypan-blue (Gibco).

### Preparation of heat-killed JBK70

*V. cholerae* strain JBK70 was inoculated into LB broth from a glycerol stock and incubated at 37 °C with agitation for 24 h. Opaque cultures were centrifuged and resuspended in an equal volume (4 mL) of cell culture media corresponding to the mammalian cells used for the assay. Samples were then diluted 1:10 in cell media and heated at 95 °C for 30 min. Heat-treated samples were cooled to room temperature before application to mammalian cells. Aliquots of heat-killed JKB70 were plated on LB agar to ensure no live bacteria were present.

### Enzyme-linked immunosorbent assays (ELISA) and cytokine measurements

Cell supernatants were analyzed using ELISA for cytokine production. ELISA kits were purchased from R&D Systems and used according to the manufacturer's instructions, including IL-1$\beta$, IL-6, IL-10, TNF-$\alpha$, MCP-1, and IL-8. Cytokine levels were determined using the 4-parameter logistic regression model calculated in GraphPad Prism (v10) according to the manufacturer's instructions. For multiplex assays, samples were analyzed using the human cytokine proinflammatory focused 15-Plex Discovery Assay (HDF15) (Eve Technologies Corporation). Briefly, samples stored at −80 °C were thawed and centrifuged at 3000×$g$ for 10 min at 4 °C. The supernatants of the samples were collected and shipped to Eve Technologies on dry ice for processing.

### PBMC stimulation and treatment with bacterial lipids and JBK70

Frozen PBMCs were thawed and plated into 24-well tissue-culture treated plates using $1 \times 10^6$ cells/well in RPMI media containing 50 ng/mL PMA. Cells were incubated for 24 h. Adherent cells were rinsed with PBS and then used for lipid treatment and JBK70 stimulation as outlined above. Briefly, the adherent cells were rinsed, and bacterial lipids were applied for 24 h. Cells were then rinsed and treated with heat-killed JBK70 for a final dilution of 1:100. After 18 h, the supernatant was collected and tested for cytokines.

### RNA extraction and sequencing

THP-1 cells were collected after exposure to *V. cholerae* and/or bacterial sphingolipids for RNA sequencing. After collecting the cell supernatant for cytokine analysis, adherent cells were treated with TRIzol (Invitrogen) and transferred into a 1.5 mL microtube. Chloroform was added, centrifuged at 4 °C, and the aqueous layer was transferred into a new 1.5 mL tube and treated with 70% ethanol (Spectrum). Samples were then processed using the Qiagen RNeasy kit (Qiagen) according to the manufacturer's instructions with on-column DNase treatment. RNA was eluted using 30 μL molecular-grade water and additionally treated with TURBO DNase (Ambion). RNA quality was assessed using TapeStation (Agilent). RNA libraries were prepared for sequencing using Illumina Stranded Total RNA Prep with Ligation with Ribo-Zero Plus according to the manufacturer's instructions. Sequencing was performed using P3 Reagents (Illumina) and RNA UD Indexes Set A, Ligation (Illumina) on a NextSeq 2000 as 150-bp pair-ended reads.

### RNAseq preprocessing and analysis

Fastq files were obtained, and RNA reads were curated and aligned using the RNA snakemake pipeline SEAsnake[106]. The raw data can be accessed through NCBI, BioProject PRJNA1170288. Briefly, RNA quality was assessed with FastQC, adapters were removed, and low-quality sequences were filtered using AdapterRemoval. Reads were aligned to the Homo sapiens genome GRCh38 (release 108) using STAR. Alignments were quality checked and filtered using samtools and Picard, and read counts were determined using Subread. RNA samples were then quality assessed and removed if libraries had <1,000,000 sequences, greater than 1 coefficient of variation (CV) coverage, and less than 75% alignment to the genome. No sample or library was removed using these criteria. Low-abundance genes with a minimum CPU of 0.1 and present in a minimum of 2 samples were filtered out. 4244 (21.22%) of 20001 genes were removed. Gene counts were then normalized by trimmed mean of M (TTM) and voom (log2 transformation of counts per million (CPM)). Analysis was then performed on the following comparisons: SL-containing with and without HK JBK70, SL-reduced with and without HK JBK70, SL-containing and SL-reduced without HK JBK70, and SL-containing and SL-reduced with HK JBK70. Linear modeling was performed using kmFit (Kimma software on R; [https://github.com/BIGslu/kimma][107] with weighted factors to obtain the estimate (log2 fold change) and FDR values. Genes were considered significant with FDR < 0.3 and log2FC greater than 1 or less than -1. Enrichment pathways were determined using the R package BIGslu/SEARchways [https://github.com/BIGslu/SEARchways] with BIGprofiler that employs clusterProfiler using the gene set database and hallmark gene sets available in the Broad Molecular Signatures Database (MSigDB)[108]. Pathways were considered significant with FDR < 0.05.

### Statistical analysis

Visualizations, figures, and statistical testing were generated using R (v4.0.2) and GraphPad Prism (v10) unless otherwise specified. Strain trees were generated using GToTree (v1.5.51)[109] and visualized using the Interactive Tree of Life (v6.1.2)[110]. Shannon indices were compared between sexes using a two-tailed Mann–Whitney $U$-test. A Kruskal–Wallis test with Dunn's adjustment for multiple comparisons was used to assess differences in Shannon index between age groups, as well as between responders and nonresponders for each vaccine response measure.

## Ethics approval and consent for publication

This study was approved by the Research Review and Ethics Review Committee of the icddr,b, in Dhaka, Bangladesh. Approvals for this work were also granted by the Institutional Review Board of the Massachusetts General Hospital and the University of Washington. Written informed consent was obtained for all participants in this study. For children below 17 years of age, permission of at least one parent/guardian was required for participation. For those aged 11–17 years of age, participant written assent was also obtained.

## Reporting summary

Further information on research design is available in the Nature Portfolio Reporting Summary linked to this article.

## Data availability

The whole-genome sequencing of the microbiome data generated in this study has been deposited in the SRA database under accession code PRJNA782606. Metadata for participant samples is provided as Supplementary Data 6. The RNA sequencing data of the THP-1-derived macrophages generated for this study have been deposited in the SRA database under accession code PRJNA1170288. The targeted quantitative mass-spectrometry-based lipidomics data of the fecal samples generated in this study have been deposited in the MassIVE repository under accession code MSV000099936 [https://doi.org/10.25345/C54J0B99S]. Metadata and results for the fecal lipidomics data are provided as Supplementary Data 7. The lipidomics data on the *Bacteroides xylanisolvens* culture generated in this study have been deposited in the MassIVE repository under MSV000099874 [https://doi.org/10.25345/C58K7592K]. Source data are provided with this paper.

## Code availability

The formula for calculating priority score is described in the methods section, Development of CAG-based strain identification method. The scripts used for priority score can be found at [https://github.com/letsgetthisfred/Dukoral_gene_level_analysis/blob/main/Generic%20Code%20for%20priority%20score%20and%20geneshot%20output.R]. Code for geneshot_extract_gene_abund.py is available at [https://gist.github.com/sminot/cebfbd84d57406b5b41b2eebffb1789f]. The code for RNA-sequencing analysis using the RNA snakemake pipeline SEAsnake v1.1 is available at [https://zenodo.org/records/11646755].

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

## Acknowledgements

We would like to thank Dr. Stephen Salipante, Dr. Kim Dill-McFarland, and the Microbial Interactions & Microbiome Center (mim_c) at the University of Washington for DNA and RNA sequencing assistance, and Dr. Thomas Hawn and Dr. Jim Kaper for laboratory resources. We thank Dr. Daniel Raftery, Dr. Danijel Djukovic, Vadim Pascua, and Maria Partida-Aguilar for the lipidomics analysis contributions. We thank Dr. Owen Jensen, Dr. Meti Debela, Chelsea Dunmire, Dr. Tsoni Peled, and Brian and Kevin Coy of Coy Labs for technical assistance. We acknowledge the study participants and the icddr,b laboratory, field, and data management staff. The funding bodies had no role in the design of the study, data collection, analysis, interpretation of data, or writing of the manuscript. This work was supported by the NIH [NIAID R01 AI103055 to J.B.H. and R.C.L.; R01 AI099243 to J.B.H. and F.Q., K08 AI123494 to A.A.W.; R01 AI106878 to E.T.R.; T32HD007233 to D.C.; R01 AI AI136979 to L.X., and NIGMS R35 GM133420 (support of S.S.M, PI: Amy D. Willis)], Fogarty International Center [D43 TW005572 to T.R.B, and K43 TW010362 to T.R.B], the University of Washington [Chief of Medicine Award and Royalty Research Foundation support to A.A.W.], the Government of the People's Republic of Bangladesh (to the International Centre for Diarrheal Disease Research [icddr,b]), and the Global Affairs Canada (to the icddr,b).

## Author contributions

D.C. contributed to methodology, formal analysis and investigation, data curation, visualization, and drafting of the original manuscript. F.J.H. contributed to methodology, code development, formal analysis and investigation, visualization, and drafting of the original manuscript. H.A.B., A.A., and M.H.K. contributed to the investigation and drafting of the original manuscript. F.C., T.R.B., A.I.K., P.C.K., and P.D. contributed to the investigation, data curation, and editing of the manuscript. S.M.M., E.P., M.G.D., and A.R. contributed to the experimental investigation. R.C.L., E.T.R., and J.B.H. contributed to the conceptualization of the project and editing of the manuscript. F.Q. contributed to the conceptualization of the project, investigation, data curation, and editing of the manuscript. L.X. contributed to the methodology, formal analysis, and editing of the original manuscript. S.S.M. contributed to the methodology, code development, formal analysis, and drafting/editing of the original manuscript. A.A.W. contributed to the conceptualization of the project, formal analysis, and drafting/editing of the original manuscript.

## Competing interests

The authors declare no competing interests.
