## [Transparent Peer Review file · Nature Communications]

Gut bacteria-derived sphingolipids alter innate immune responses to oral cholera vaccine antigens

Corresponding Author: Dr Ana Weil

Version 0:

Reviewer comments:

Reviewer #1

(Remarks to the Author)

Weil and colleagues present an interesting premise that variations in response to the oral cholera vaccine is influenced by gut microbial sphingolipids. Authors present a set of in vitro experiments to suggest that sphingolipid-producing bacteria may directly contribute to vaccine response through the interaction of bacterial sphingolipids with immune cells. Major concerns with the manuscript are in the strength of the assertions that gut microbiome derived sphingolipids are involved in vaccine responsiveness through mediating innate immunity. In all the bacterial work, the attribution of response was based on lipid extracts derived from treating *Bacteroides* with a drug that inhibits de novo sphingolipid synthesis, yet no lipid profiling of these extracts was presented to determine the level of inhibition achieved. Moreover, if events are to be attributed to bacterial sphingolipids, then it is preferable that these sphingolipids are isolated, identified, and then used to assay the system. Other concerns include the strength of the connection of the responder phenotype to microbiome community composition and the premise for the focus on sphingolipids as a driver of the response. Overall, the responses to bacterial lipid extracts are interesting but more details on the lipid profiles and a stronger connection to the human cohort study would be needed to support a meaningful role of gut bacterial-derived sphingolipids on oral cholera vaccine response. Below are specific comments on the manuscript.

1. How was it determined that the myriocin treated Bx cells were sphingolipid free? Would need to measure the lipids in the sphingolipid fractions in these experiments to make stronger statements.
2. Differing levels of mainly host-derived sphingolipids in stool suggest that there may be other non-microbiome related intestinal track/lipid processing characteristics between the groups.
3. Were the sphingolipid profiles measured in Bx present in the lipids in responder stool?
4. What were the other lipids that were high in responders – maybe they are more responsible for phenotype
5. Is there any cell death associated with lipid treatment in cell culture model?

Minor:

1. Line 490 *Bacteroides* serine palmitoyltransferase (spt) abundance was determined by querying BT_0780. Perhaps this is meant to be BT_0870. BT_0780 is an uncharacterized acetyltransferase.
2. Figure 2C – color scale not showing up in provided version
3. SPT gene abundance is almost a direct readout of presence of *Bacteroides*/*Prevotella* species – any insight into why this enrichment didn't come out in the community composition analysis?

(Remarks on code availability)

Reviewer #2

(Remarks to the Author)

The manuscript entitled "Gut bacteria-derived sphingolipids alter innate immune responses to oral cholera vaccine antigens" is a follow up study from the parent mSphere article entitled "Cholera toxin and O-specific polysaccharide immune responses after oral cholera vaccination with Dukoral in different age groups of Bangladeshi participants". Here the authors utilise the faecal materials collected at the time of vaccination and undertake metagenomic sequencing to infer microbiome associations to the cholera toxin vaccine responder and non-responders.

The authors subsequently undertake a fundamental unit of analysis assessment of the metagenomic data. Using this approach, the authors find an association between *Bacteroides*, specifically *xylanisolvans* strains and *Prevotella* and memory B cells- which appears to be strain specific.

Next the authors undertake a quantitative mass spectrometry-based lipidomics approach to the responder/non-responder stool samples. They find that several sphingolipid derivatives were more abundant in vaccine responders.

Subsequently the authors undertake a series of experiments to validate these findings in human macrophages. They choose a particular *Bacteroides xylanisolvans*, isolated from one of the participants, for these studies.

Comments

The article is well written and easy to understand. It details the metagenomic sequencing of 98 participants in 3 defined cohorts between 2-5 years, 5-17 years, and 18-46 years. The authors clearly state the male/female ratios and the first 2 cohorts are equally balanced, with only the final cohort having a female bias.

As the parent article is published elsewhere, the authors are unfortunately unable to detail how the responder/non-responder ratios were defined. Instead the authors focus directly on the metagenomics data. Their findings indicate a possible difference in microbiomal diversity by sex; however, they do not find a significant difference by Shannon diversity or Bray-Curtis community structure between the vaccine responders and non-responders and hence.

The reviewer also has reservations regarding the THP-1 validation of these findings. This is the most important part of this manuscript, as it attempts to validate this important association. The authors take a series of experiments using a mammalian cell line, THP-1, when they should be using human cells derived from PMBCs. The initial assessment of innate responses includes ELISA validation of IL-1b, IL-6, TNF-a and IL-8 (figure 4). However, the authors do go on to expand this model (figure 5) and undertake a more thorough analysis using a 15-plex multi assessment of cytokines and RNA sequencing of resultant THP-1 cells. However, this assessment should be followed by macrophages derived from participant/control PMBCs and include responder material if possible and or in vivo validation in mouse models. Currently this association is not sufficiently validated.

Methods:

Authors need to detail fecal sample collection and timing information.

Authors do not detail the library preparation process at all.

Figure 1: Inclusion of schematic figure detailing sample collection timepoints and participant cohorts would be a useful inclusion

Figure 2:2c- Missing priority scoring colour information.

Figure 3: Stats are not shown on the graph

(Remarks on code availability)

Reviewer #3

(Remarks to the Author)

Chac et al. present a study investigating mechanisms by which gut microbiota may influence response to an oral cholera vaccine.

The first half of the study focusses on clustering de novo assembled bacterial genes into co-abundant gene groups (CAGs), which are subsequently correlated with several measures of humoral immune response to vaccination in a cohort of 89 individuals.

The second half of the study focusses on exploring the impact of microbially-produced sphingolipids on innate immune responses through in vitro stimulation of THP-1 derived macrophages with heat-killed *V. cholerae* strain JBK7.

While both of these halves present interesting and potentially important work, I don't see strong justification for combining them into a single manuscript. In the first half of the manuscript, much emphasis is justifiably placed on de novo functional and strain-level characterization of the microbiome in relation to vaccine response. However, the only reason given for focussing on sphingolipids in the second half of the study is that genera identified as containing CAGs correlating with vaccine response (e.g. *Bacteroides* and *Prevotella* - lines 181-182) are known to produce sphingolipids.

Critically, there seems to be no evidence that any of the CAGs identified as correlating with vaccine-induced IgA/IgG are associated with production of sphingolipids. There is also no effort to identify evidence of enrichment in sphingolipid production potential in strains identified as important using the humoral CAG-based strain identification method.

Without any bioinformatic evidence for sphingolipids relating to humoral vaccine response it is unclear a) why a metagenomic shotgun sequencing and complex CAG-based analysis approach is necessary in order to identify sphingolipid producing genera, and b) why the study subsequently chose to focus on sphingolipids rather than follow up on functions

encoded in any of the 323 CAGs that were identified as correlating with vaccine response.

Unfortunately, without clearer evidence to justify linking the two halves of this study I can find no rationale for publishing it as it's currently presented. I recognise the potential importance of this study as well as the careful and innovative work that's been done to date, and am including additional minor comments below in the hope that they are helpful towards a future publication.

MINOR COMMENTS:

LINE 396: Would be very useful to clarify if stool samples were collected on site, or whether they were collected at home and brought to the study centre? How long were they out of the freezer?

LINE 90: Use of the word simple in section header is subjective.

LINE 98: Serotypes (Ogawa and Inaba) are introduced here without prior explanation. A small amount of detail for unfamiliar readers would be very helpful.

LINE 414: Given the emphasis on CAG-based strain identification (lines 148, 457), it would be useful to see more justification of the approach taken for taxonomic annotation. In particular that use of translated alignment and non-redundant gene clustering (90% sequence identity) are not limiting/skewing the accuracy of taxonomic identification. Why not just assign taxonomy to the original contigs generated by Megahit, thereby avoiding multiple unnecessary levels of nucleotide-protein translation that are likely to reduce the accuracy of this important step?

LINE 100: It's unclear whether the result reporting differences in IgA vs IgG MBC response are general, or referring to only those individuals where an increase in MBC response was observed (line 99).

LINE 101: I can't easily discern this result from the data presented (Fig 1A?) and more explanation would be helpful. i) why are the numbers of OSP-associated MBC responders and non-responders different between IgA vs IgG comparisons (62 vs 46, respectively)? ii) what evidence is there that the numbers developing an IgG directed OSP-specific MBC response (23) is greater than the number developing an IgA directed OSP-specific MBC response (19)?

LINE 112: Given 22/89 study participants are aged 2-5, this is surprising and looks to be driven by a subset of individuals aged 20+ with low diversity scores. Could the authors not provide rarefaction/collectors curves to demonstrate all individuals were sequenced to an adequate depth to ensure robust diversity measures?

LINE 450-453: A bit more detail would be helpful. How were read counts divided for non-redundant genes that aligned to multiple reference genomes? Were summed counts adjusted to account for variation in reference genome size?

FIGURE S3: It would be very useful to see how, on these plots, the number of genomes aligned to relates to the number of genes in a CAG.

LINE 153: Given the emphasis on strain-specificity, it would be useful to provide a little more detail/i) What is the working definition of a strain within the RefSeq database used to map CAGs, ii) how does the strain identification method (Line 457) account for differences in variation in the number of reference genomes across taxonomic groups, iii) what evidence is there that differential alignment success of CAGs to genomes isn't simply an artefact of the variable quality of the underlying genome assemblies?

LINE 229-233: I found this sentence difficult to understand as it appears to contrast SL-containing Bx supernatant to SL-containing lipid fractions, which doesn't accord with the referenced figure panel (Fig 5A).

LINE 239: Incorrect figure reference?

LINE 287: I disagree with the statement – here and elsewhere – that the approach is reference free. Unless I've misunderstood, strain annotation of CAGs is dependent on mapping them to genomes in the RefSeq database?

(Remarks on code availability)

Version 1:

Reviewer comments:

Reviewer #2

(Remarks to the Author)

The authors have addressed my comments on those of the other reviewers in depth. I have no further comments to make.

(Remarks on code availability)

REVIEWER COMMENTS

Reviewer #1 (Remarks to the Author):

Weil and colleagues present an interesting premise that variations in response to the oral cholera vaccine is influenced by gut microbial sphingolipids. Authors present a set of in vitro experiments to suggest that sphingolipid-producing bacteria may directly contribute to vaccine response through the interaction of bacterial sphingolipids with immune cells. Major concerns with the manuscript are in the strength of the assertions that gut microbiome derived sphingolipids are involved in vaccine responsiveness through mediating innate immunity. In all the bacterial work, the attribution of response was based on lipid extracts derived from treating *Bacteroides* with a drug that inhibits de novo sphingolipid synthesis, yet no lipid profiling of these extracts was presented to determine the level of inhibition achieved. Moreover, if events are to be attributed to bacterial sphingolipids, then it is preferable that these sphingolipids are isolated, identified, and then used to assay the system. Other concerns include the strength of the connection of the responder phenotype to microbiome community composition and the premise for the focus on sphingolipids as a driver of the response. Overall, the responses to bacterial lipid extracts are interesting but more details on the lipid profiles and a stronger connection to the human cohort study would be needed to support a meaningful role of gut bacterial-derived sphingolipids on oral cholera vaccine response. Below are specific comments on the manuscript.

Overall Response:

We appreciate this critique provided to help us improve our manuscript. To address the reviewer concerns about the overall strength of assertions that bacterial-derived sphingolipids impact our model of vaccine responsiveness, we have performed additional experiments and added to our collaborative group Dr. Libin Xu (bacterial lipid expert from our Department of Medicinal Chemistry at the University of Washington) who has performed ion mobility-mass spectrometry to determine the lipids present in bacterial supernatant from our lipid producing bacteria. These additional experiments, in addition to other experimental additions and explanations below, have strengthened the assertions we present in this manuscript.

Responses:

1. How was it determined that the myriocin treated [*Bacteroides xyloxylosum*] Bx cells were sphingolipid free? Would need to measure the lipids in the sphingolipid fractions in these experiments to make stronger statements.

- **Response:** We agree that this additional analysis will strengthen our manuscript. To address this, we measured the lipids in myriocin-treated *B. xyloxylosum* culture lipid extracts using hydrophilic interaction liquid chromatography (HILIC)

followed by ion mobility-mass spectrometry (IM-MS). We partnered with Dr. Libin Xu, a medical chemist and expert in bacterial-derived lipids for this work and for the interpretation of the results. The methods used to extract lipids from Bx cells were identical to those used in our cell culture experiments. In HILIC-IM-MS, ceramides are the primary readout for estimating sphingolipid abundance. Ceramides are an essential structural backbone of all sphingolipids and are measured to approximate sphingolipid content. The use of myriocin inhibits *spt* (serine palmitoyltransferase) function, and this is the first step required for bacterial *de novo* sphingolipid synthesis from a ceramide.

The total ceramides we measured in myriocin-treated Bx lipids fractions were significantly lower than in non-myriocin treated lipid fractions. These results were added the manuscript as **Supplemental Figure S7**. In these results, we observed that the *Bacteroides* lipid fraction has reduced, but not zero, ceramides. Therefore, we have revised the text to refer to the myriocin-treated samples as sphingolipid-reduced or SL-reduced (rather than sphingolipid-free, as in our initial submission).

Figure S7. Myriocin treatment of *Bacteroides xylanisolvens* culture lipid extract reduces ceramide levels.

B. xylanisolvens (Bx) was grown in BHI supplemented media in anaerobic conditions with or without the *spt* inhibitor, myriocin. Lipids were then extracted using the Bligh-Dyer and measured using hydrophilic interaction liquid chromatography and ion mobility-mass spectrometry. Data is shown as the total intensity of detected positive (Cer+) and negative (Cer-) mode ceramides tested in triplicate. Intensity refers to the signal strength of an ion detected by the mass spectrometer and is proportional to the abundance of that ion. Ions are measured in either negative or positive mode based on their chemical structure and functional groups. Unpaired t tests with SEM are shown. ***, $P \leq 0.001$.

2. Differing levels of mainly host-derived sphingolipids in stool suggest that there may

be other non-microbiome related intestinal track/lipid processing characteristics between the groups.

- **Response:** We agree and acknowledge that the sphingolipids found in human stool may be host derived, diet derived, or bacterial derived. Only some sphingolipid structures can be determined to be derived from bacteria based on branching structure. To our knowledge, it is currently not possible to determine the origin of some stool lipids, because *Bacteroides* species produce some similar sphingolipid structures to those derived from humans (this is described in An et al., 2010 - PMID: 20855611; Johnson et al., 2020 - PMID: 32424203) and because plant materials and other dietary sources may also contain non-human sphingolipids. We have expanded on our limitations to include that host-derived sphingolipids in stool, that are unrelated to the microbiota, may also be responsible for the findings we observed in human stool.

3. Were the sphingolipid profiles measured in Bx present in the lipids in responder stool?

- **Response:** To address this important comment, and further link our human data with our experimental results, we sought to compare the lipid profiles from study participant stool and the *B. xylanisolvans*-produced lipids. Due to limitations of branching and structural resolution using ion mobility-mass spectrometry and incomplete scientific knowledge about sphingolipid structures (especially those made by bacteria), our confidence in fully resolving structures of even some lipids contained in stool was extremely low. However, in our analyses of the stool and bacterial lipids, we identified two bacterial-derived sphingolipids detected with high confidence both in stool and in the bacterial lipid culture. **These two identifiable ceramides were more abundant in vaccine responders and also found in *B. xylanisolvans* lipids – these are CER(16:0) and CER(18:1).** We have revised the manuscript accordingly and added the supplementary data **Table S9** that shows the abundances of these two lipids in extracts from *Bacteroides xylanisolvans* lipid fractions. Study participant stool measures of these two ceramides are also shown in the context of other lipids detected in human stool in Figure 3B, and quantities are listed in Table S8. To illustrate this point and the comparison, we have created a hybrid table here for this reviewer question.

	Bacterial lipidomics (Average of biological triplicates)			Human Fecal lipidomics		
	SL-containing mean±SD	SL-reduced mean±SD	Unpaired T test P-value	Responder (N=9) average in nmol/g	Nonresponder (N=7) average in nmol/g	Multiple T testing P-value
Cer(d18:1/18:1) [M+H] ⁺	2737 (±907.6)	1292 (±208.8)	0.054	4.44	0.93	0.016
Cer(d18:1/16:0) [M-H] ⁻	959.4 (±563.7)	144.8 (±23.9)	0.066	38.9	11.1	0.008

Table S9 and excerpt of Table S8, combined data: Ceramides detected in *B. xylanisolvans* lipid fractions with or without *spt* inhibitor myriocin and in study participant stool. The SL-reduced condition in bacterial lipidomics consist of *B.*

xylanisolvans grown in culture with the *spt* inhibitor, myriocin. Lipids were the extracted from culture using the Bligh-Dyer method and measured by ion mobility-mass spectrometry. Fecal lipids from study participants were measured using the Sciex Lipidyzer platform described in the methods and as in PMID: 31832778.

4. What were the other lipids that were high in responders – maybe they are more responsible for phenotype

- **Response:** The other lipids that were differentially abundant in vaccine responders include diacylglycerol, cholesterol esters, and free fatty acids. The most abundant lipids and the association with responder and nonresponder status and abundances can be found in our new **Table S8**. We have also added comment on this to the manuscript and study limitations. These lipid species are primarily produced by the host and were not further investigated given our focus on the microbiome and microbiome-derived metabolites.

5. Is there any cell death associated with lipid treatment in cell culture model?

- **Response:** We assessed the cell numbers for 24 hours and did not observe any cell death associated with the lipid treatment in our THP-1 model. This data has been added to the manuscript as **Figure S9**.

Figure S9. Lipid treatment did not alter THP-1 cell viability. The number of adherent cells were counted after THP-1 derived macrophages were incubated with *Bacteroides xylanisolvans* lipid extracts that was grown with or without myriocin, SL-reduced and SL-containing, respectively, for 24 hours. Statistical analysis was performed using one-way ANOVA. ns, $P > 0.05$. Bars indicate mean with SEM.

Minor:

1. Line 490 *Bacteroides* serine palmitoyltransferase (*spt*) abundance was determined by querying BT_0780. Perhaps this is meant to be BT_0870. BT_0780 is an uncharacterized acetyltransferase.

- **Response:** This is a typo, and the correct gene is BT_0870; this has been corrected. Thank you for this important correction.

2. Figure 2C – color scale not showing up in provided version

- **Response:** The color scale for Figure 2C has been revised.

3. SPT gene abundance is almost a direct readout of presence of *Bacteroides*/*Prevotella* species – any insight into why this enrichment didn't come out in the community composition analysis?

- **Response:** The primary reason that these genera did not emerge from our computational analysis is because not all the species in the *Bacteroides* and *Prevotella* genera have the *spt* gene. *Spt* is strain specific and is also not consistent across one species within these genera. To exemplify this, in Figure 2d we show a phylogenetic tree of *Bacteroides xyloxylicans* strains found in our study population, and only 7 of the 21 *Bacteroides xyloxylicans* strains contained *spt*. Therefore, due to the abundance of *spt* negative *Bacteroides* strains, our study was not powered enough to detect this difference at the genus or species level. In particular, the *Prevotella* genus is appreciated to encompass an extremely wide genetic and functional diversity (reviewed in PMID: 34050328). Therefore, while we did not examine this genus as closely in this manuscript, the *Prevotella* detected in our study population are also very likely to have a mix of *spt*+ and some *spt*- strains, similar to our findings in *Bacteroides*.

Reviewer #2 (Remarks to the Author):

The manuscript entitled "Gut bacteria-derived sphingolipids alter innate immune responses to oral cholera vaccine antigens" is a follow up study from the parent mSphere article entitled "Cholera toxin and O-specific polysaccharide immune responses after oral cholera vaccination with Dukoral in different age groups of Bangladeshi participants". Here the authors utilise the faecal materials collected at the time of vaccination and undertake metagenomic sequencing to infer microbiome associations to the cholera toxin vaccine responder and non-responders.

The authors subsequently undertake a fundamental unit of analysis assessment of the metagenomic data. Using this approach, the authors find an association between *Bacteroides*, specifically *xylanisolvans* strains and *Prevotella* and memory B cells- which appears to be strain specific.

Next the authors undertake a quantitative mass spectrometry-based lipidomics approach to the responder/non-responder stool samples. They find that several sphingolipid derivatives were more abundant in vaccine responders.

Subsequently the authors undertake a series of experiments to validate these findings in human macrophages. They choose a particular *Bacteroides xylanisolvans*, isolated from one of the participants, for these studies.

Comments

The article is well written and easy to understand. It details the metagenomic sequencing of 98 participants in 3 defined cohorts between 2-5 years, 5-17 years, and 18-46 years. The authors clearly state the male/female ratios and the first 2 cohorts are equally balanced, with only the final cohort having a female bias.

As the parent article is published elsewhere, the authors are unfortunately unable to detail how the responder/non-responder ratios were defined. Instead the authors focus directly on the metagenomics data. Their findings indicate a possible difference in microbiomal diversity by sex; however, they do not find a significant difference by Shannon diversity or Bray-Curtis community structure between the vaccine responders and non-responders and hence.

The reviewer also has reservations regarding the THP-1 validation of these findings. This is the most important part of this manuscript, as it attempts to validate this important association. The authors take a series of experiments using a mammalian cell line, THP-1, when they should be using human cells derived from PMBCs. The initial assessment of innate responses includes ELISA validation of IL-1b, IL-6, TNF-a and IL-8 (figure 4). However, the authors do go on to expand this model (figure 5) and undertake a more thorough analysis using a 15-plex multi assessment of cytokines and RNA sequencing of resultant THP-1 cells. However, this assessment should be followed by macrophages derived from participant/control PMBCs and include responder material if possible and or in vivo validation in mouse models. Currently this association is not

sufficiently validated.

Overall Response:

We appreciate this reviewer's evaluation of our manuscript. First, we have clarified in the manuscript the immunologic measures we used and the definitions of vaccine recipients as responders or nonresponders. Vaccine recipients were grouped according to development of cholera toxin- and O-specific polysaccharide-specific memory B cell (MBC) response measures in peripheral blood, developed between the pre- and post-vaccination timepoints. Both IgA and IgG-specific responses for each of these antigens were measured in each participant. Individuals with increased MBC counts following vaccination were defined as vaccine responders (R) while those that had unchanged or decreased MBC response after vaccination were defined as vaccine nonresponders (NR). We have revised the manuscript with a clearer description of this information to address this reviewer comment.

We agree that validation of our *in vitro* human macrophage experiments is needed, and we wanted to address this using more diverse human immune cells. We used peripheral blood mononuclear cells (PBMCs) donated by healthy adults. We repeated our THP-1 experiment using PBMCs; specifically, we used pre-treatment of cells with *B. xylanisolvans* lipids and subsequent heat-killed JBK70 stimulation. We then measure innate cytokine responses and found increased IL-6 across PBMC donors between the SL-containing lipids condition compared to SL-reduced lipids. Although we did find donor-to-donor variability in the results, we overall found that PBMC experiments across four donors validated our results from human macrophages and we have added this data to the manuscript as a supplemental figure: **Figure S14**.

Figure S14. Stimulated peripheral blood mononuclear cells (PBMCs) have increased inflammatory cytokine responses to JBK70 following SL-containing lipid pretreatment.

Fold change in cytokine response in supernatant from PBMCs after preconditioning with *B. xylanisolvans* lipid extracts and treated with heat-killed JBK70. Bars represent mean with SEM. Unpaired T-test comparing fold change: *, $P < 0.05$; **, $P < 0.01$; ***, $P < 0.001$. Each number represents a different PBMC donor. #1 = 31 yr female; #2 = 25 yr male; #3 = 36 yr female; and #4 = 35 yr

male. Three to four technical replicates were performed on each experimental condition depending on the PBMCs available after thawing.

For the validation experiment, we used human-derived models rather than mouse models because *V. cholerae* infection is human-restricted and there is not a suitable model for modeling oral cholera vaccine responsiveness. Studies have been done using mice, but these often use artificial routes of antigen delivery and poorly predict human immune responses. Mouse models for immune response to oral vaccines also do not model the human microbiota, and precise experimental manipulation of the gut environment and *in situ* analysis of host-microbe interactions is very challenging in animals. For these reasons, we pursued validation studies in human-derived models rather than in animals. Unfortunately, additional samples from study participants such as peripheral blood or responder material (fecal samples) were not available for further validation assays.

Response to questions and minor comments:

Methods:

Authors need to detail fecal sample collection and timing information.

- **Response:** The collections and timing of fecal samples have been revised regarding sample collection and timing, and this is listed in the “Study design, subject enrollment, and sample collection” section of the methods, and shown in a revised **Figure 1a**.

Authors do not detail the library preparation process at all.

- **Response:** We have added details to the library preparation in the revised methods section of the manuscript.

Figures:

Figure 1: Inclusion of schematic figure detailing sample collection timepoints and participant cohorts would be a useful inclusion

- **Response:** A **schematic was added as Figure 1a** to illustrate and clarify sample collection timepoints, the vaccination schedule.

Excerpt of Figure 1a. Schematic of study design, OCV dosing, and sample collection timepoints, adapted from parent study. Responders were defined as an OCV recipient who developed an increase in MBC response between pre- and post-vaccination timepoints, and nonresponders were defined as participants with unchanged or decreased MBC responses between timepoints.

Figure 2:2c- Missing priority scoring colour information.

- **Response:** The color scale for Figure 2C has been corrected in the revised manuscript.

Figure 3: Stats are not shown on the graph

- **Response:** The figure has been corrected in the revised manuscript.

Reviewer #3 (Remarks to the Author):

Chac et al. present a study investigating mechanisms by which gut microbiota may influence response to an oral cholera vaccine.

The first half of the study focusses on clustering de novo assembled bacterial genes into co-abundant gene groups (CAGs), which are subsequently correlated with several measures of humoral immune response to vaccination a cohort of 89 individuals.

The second half of the study focusses on exploring the impact of microbially-produced sphingolipids on innate immune responses through in vitro stimulation of THP-1 derived macrophages with heat-killed *V. cholerae* strain JBK7.

While both of these halves present interesting and potentially important work, I don't see strong justification for combining them into a single manuscript. In the first half of the manuscript, much emphasis is justifiably placed on de novo functional and strain-level characterization of the microbiome in relation to vaccine response. However, the only reason given for focussing on sphingolipids in the second half of the study is that genera identified as containing CAGs correlating with vaccine response (e.g. *Bacteroides* and *Prevotella* - lines 181-182) are known to produce sphingolipids.

Critically, there seems to be no evidence that any of the CAGs identified as correlating with vaccine-induced IgA/IgG are associated with production of sphingolipids. There is also no effort to identify evidence of enrichment in sphingolipid production potential in strains identified as important using the humoral CAG-based strain identification method.

Without any bioinformatic evidence for sphingolipids relating to humoral vaccine response it is unclear a) why a metagenomic shotgun sequencing and complex CAG-based analysis approach is necessary in order to identify sphingolipid producing genera, and b) why the study subsequently chose to focus on sphingolipids rather than follow up on functions encoded in any of the 323 CAGs that were identified as correlating with vaccine response.

Unfortunately, without clearer evidence to justify linking the two halves of this study I can find no rationale for publishing it as it's currently presented. I recognise the potential importance of this study as well as the careful and innovative work that's been done to date, and am including additional minor comments below in the hope that they are helpful towards a future publication.

Overall Response:

Thank you for this assessment of the structure of our study and description of a disconnect between two "halves" of work. Our overall goal in presenting this work as

a whole and to illustrate a full computational to experimental testing pipeline, a process that starts with human samples and progresses to exploring mechanisms using *in vitro* models. Thus, we acknowledge this paper contains a range of methods and results, and we feel there is inherent value in presenting translational science such as this in a single manuscript. To achieve a smoother and more linear presentation of this process, we have reorganized and significantly revised the manuscript to clarify the sequential steps in this work that led to each next phase. With these changes, we feel this version more accurately reflects the series of steps we undertook as the study progressed, and this unites the two “halves” by demonstrating more clearly our rationale. We have added data to illustrate the steps that led to choosing specific *in vitro* experiments, based on the computational analysis, and we feel this has helped to unite the two “halves” of the manuscript.

Specifically, we have added:

- Serine palmitoyltransferase gene(*spt*) counts from the stool of Responder and Nonresponder study participants. This is the enzyme required for bacteria to synthesize sphingolipids. *Spt* was more abundant in the stool of vaccine responders (**Figure 3a**), and this finding then resulted in our selection of a *spt*-positive *B. xylanisolvans* strain for experimental testing. We have reordered the description of results to demonstrate that the specific genera highlighted by our metagenomic results (sphingolipid-producing microbes) led us to next quantify *spt* in study participant stool.
- We examined lipid content and abundance in study participant stool and compared this to the culture supernatant of lipids produced by *Bacteroides xylanisolvans*, and found specific ceramide groups (the head group required in bacterial sphingolipid synthesis) that were differentially abundant in these two groups, described in Table S8 and new **Table S9** (link between these areas is further detailed in the revised manuscript and in the explanation to point #3 raised by Reviewer #1).
- Revisions to the manuscript to reflect that the CAG analysis did not single out in the *Bacteroides* and *Prevotella* genera as associated with vaccine responsiveness because not all the species in the *Bacteroides* and *Prevotella* genera have the *spt* gene. ***Spt* is strain specific** and is thus not consistent across one species within these genera. To exemplify this, we show in Figure 2D a phylogenetic tree of *Bacteroides xylanisolvans* strains found in our study population, and only 7 of the 21 *Bacteroides xylanisolvans* strains contained *spt*. Therefore, due to the abundance of *spt* negative *Bacteroides* strains, our study was not powered enough to detect this difference at the genus or species level.
- A clarification that CAGs are identified in this study for the purpose of mapping to genomes for the identification of specific strains for experimental testing. It would have been ideal if CAGs could be considered as functional units that we could query experimentally, but, unfortunately, a query of functions encoded by a CAG is not possible because all CAGs contain multiple genes, most of which are unannotated or nonspecific (Table S2).

We have revised the manuscript on all of the above points, and appreciate this reviewer's comments about the disjointedness of our initial draft. This review has helped us to significantly improve the manuscript, in our opinion.

Response to questions and comments:

MINOR COMMENTS:

LINE 396: Would be very useful to clarify if stool samples were collected on site, or whether they were collected at home and brought to the study centre? How long were they out of the freezer?

- **Response:** The collections and timing of fecal samples have been reworded for clarity and moved to the "Study design, subject enrollment, and sample collection" in the Methods. Stool samples were collected at the International Centre for Diarrheal Disease Research, Bangladesh, by study staff on Day 0 of the study (the day of first OCV dose) and immediately and frozen at -80C, and then shipped on dry ice to US investigators for storage, DNA extraction, and fecal lipidomics. We have added a schematic as **Figure 1a** to further demonstrate our study design in the revised manuscript.

LINE 90: Use of the word simple in section header is subjective.

- **Response:** This section header has been revised in the new draft of the manuscript.

LINE 98: Serotypes (Ogawa and Inaba) are introduced here without prior explanation. A small amount of detail for unfamiliar readers would be very helpful.

- **Response:** We have revised the manuscript introduction to include a description of the Ogawa and Inaba *V. cholerae* O1 serotypes.

LINE 414: Given the emphasis on CAG-based strain identification (lines 148, 457), it would be useful to see more justification of the approach taken for taxonomic annotation. In particular that use of translated alignment and non-redundant gene clustering (90% sequence identity) are not limiting/skewing the accuracy of taxonomic identification. Why not just assign taxonomy to the original contigs generated by Megahit, thereby avoiding multiple unnecessary levels of nucleotide-protein translation that are likely to reduce the accuracy of this important step?

- **Response:** We have two primary motivations for using protein-level sequences instead of nucleotide-level contigs. (1) Using a comparison contig-level abundances across all samples would require some manner of co-assembly which can be prohibitively computationally intensive at this scale. (2) Functionally identical biological functions may be performed by genes with high amino acid sequence identity and secondary structure despite their nucleotide sequences being dissimilar enough to prevent assembly as a single contig.

LINE 100: It's unclear whether the result reporting differences in IgA vs IgG MBC response are general, or referring to only those individuals where an increase in MBC response was observed (line 99).

- We have revised the text to make clear that the summary data presented for vaccine responders refers only to study participants with increases in MBC responses after vaccination (vaccine responders).

LINE 101: I can't easily discern this result from the data presented (Fig 1A?) and more explanation would be helpful. i) why are the numbers of OSP-associated MBC responders and non-responders different between IgA vs IgG comparisons (62 vs 46, respectively)? ii) what evidence is there that the numbers developing an IgG directed OSP-specific MBC response (23) is greater than the number developing an IgA directed OSP-specific MBC response (19)?

- **Response:** (i) The denominator of participants with OSP-associated MBC IgA responses differ from those with IgG responses due to the incomplete data (missingness) for some participants in the IgG MBC measurement. (ii) For each participant, IgG and IgA antibody secreting cells were assessed separately using ELISPOT assays (these results were published in PMID: 38391226), and thus, the two groups differ. There were more individuals who had an increase in OSP-specific memory B cells IgG responses (23/46 or 50%) compared to OSP-IgA MBC responses (19/62 or 30.6%).

LINE 112: Given 22/89 study participants are aged 2-5, this is surprising and looks to be driven by a subset of individuals aged 20+ with low diversity scores. Could the authors not provide rarefaction/collectors curves to demonstrate all individuals were sequenced to an adequate depth to ensure robust diversity measures?

- **Response:** We have added a rarefaction curve to demonstrate that all samples yielded at least 600,000 reads and appear to capture the majority of observed species. This information has been added to the manuscript as **Supplemental Figure S3**.

-

Figure S3. Rarefaction curve of microbiome data. Each line represents an individual sample. Rarefaction performed on species-level identification of microbiome species.

LINE 450-453: A bit more detail would be helpful. How were read counts divided for non-redundant genes that aligned to multiple reference genomes? Were summed counts adjusted to account for variation in reference genome size?

- **Response:** We have added detail to clarify the justification for the approach used with alignment of non-redundant genes to multiple reference genomes, in which the biological expectation is that multiple representatives from a species or genus may contain homologous genes (see revised Methods section “CAG alignment to bacterial genomes”). While the summed counts were not adjusted for genome size, this would not have any impact on our statistical analysis which used the marginal association of each genome in isolation, and the genome size adjustment factor would have been applied equally to the abundance values across all samples for the same genome.

FIGURE S3: It would be very useful to see how, on these plots, the number of genomes aligned to relates to the number of genes in a CAG.

- **Response:** Figure S4 has been revised to illustrate the relationship between the number of genes in significant CAGs with the number of genomes that were aligned to that CAG.

Figure S4. Number of NCBI reference bacterial genomes mapped to CAGs that were found to be associated with vaccine responsiveness. Only significant CAGs associated with vaccine response are shown, and each is represented as a circle or X. **Significant CAGs are defined as a CAG containing ≥ 2 genes with a q value of ≤ 0.1 for an association with the listed vaccine response measure.** Red x's indicate CAGs that had no genomes aligned. The x-axis indicates the number of genes within each CAG while the y-axis represents the number of reference genomes to which genes in those CAGs align to with $>90\%$ sequence identity. Significant CAGs aligning to only one bacterial genome are not shown (20 CAGs for OSP IgA, 3 CAGs for OSP IgG, 12 CAGs for CT IgA, 5 CAGs for CT IgG).

LINE 153: Given the emphasis on strain-specificity, it would be useful to provide a little more detail/i) What is the working definition of a strain within the RefSeq database used to map CAGs, ii) how does the strain identification method (Line 457) account for differences in variation in the number of reference genomes across taxonomic groups, iii) what evidence is there that differential alignment success of CAGs to genomes isn't simply an artefact of the variable quality of the underlying genome assemblies?

- **Response:** (1) The RefSeq database does not present a formal definition of a strain, and the term is being used here as a reference to a single reference

genome which is present in the RefSeq database. We have added this definition in the methods for clarification in the revised manuscript. (2) The strain identification method is not expected to correct for differences in reference database composition, and is simply referring to the detection of genomic sequences which are homologous to the reference genome for a particular strain. (3) We have revised the limitations described in the manuscript to make clear that genomes with lower assembly quality will be more difficult to detect using this analysis approach.

LINE 229-233: I found this sentence difficult to understand as it appears to contrast SL-containing Bx supernatant to SL-containing lipid fractions, which doesn't accord with the referenced figure panel (Fig 5A).

- **Response:** The sentence has been revised for clarity in the manuscript.

LINE 239: Incorrect figure reference?

- **Response:** Thank you for this correction, and the figure references have been corrected in the revised manuscript.

LINE 287: I disagree with the statement – here and elsewhere – that the approach is reference free. Unless I've misunderstood, strain annotation of CAGs is dependent on mapping them to genomes in the RefSeq database?

- **Response:** The reference-free aspect of our study is the portion of the analysis in which CAGs are generated, quantification of the relative abundance of CAGs across metagenomes, and identifying CAGs which are differentially abundant across metagenomes as a function of vaccine response. The result of that portion of the analysis is a collection of organisms (defined by the presence of a particular collection of genes encoded in their genome) which are associated by relative abundance with vaccine response. The subsequent step of alignment to the RefSeq genome database is reference-dependent, but it is independent from and ancillary to the CAG analysis itself. We have revised this in the manuscript to clarify what we are referring to as “reference-free”.